# Equivariant Neural Operator Learning with Graphon Convolution

**Chaoran Cheng**
University of Illinois Urbana-Champaign
`chaoran7@illinois.edu`

**Jian Peng**
University of Illinois Urbana-Champaign
`jianpeng@illinois.edu`

## Abstract

We propose a general architecture that combines the coefficient learning scheme with a residual operator layer for learning mappings between continuous functions in the 3D Euclidean space. Our proposed model is guaranteed to achieve SE(3)-equivariance by design. From the graph spectrum view, our method can be interpreted as convolution on graphons (dense graphs with infinitely many nodes), which we term *InfGCN*. By leveraging both the continuous graphon structure and the discrete graph structure of the input data, our model can effectively capture the geometric information while preserving equivariance. Through extensive experiments on large-scale electron density datasets, we observed that our model significantly outperformed the current state-of-the-art architectures. Multiple ablation studies were also carried out to demonstrate the effectiveness of the proposed architecture.

## 1 Introduction

Continuous functions in the 3D Euclidean space are widely encountered in science and engineering domains, and learning the mappings between these functions has potentially an amplitude of applications. For example, the Schrödinger equation for the wave-like behavior of the electron in a molecule, the Helmholtz equation for the time-independent wave functions, and the Navier–Stokes equation for the dynamics of fluids all output a continuous function spanning over $\mathbb{R}^3$ given the initial input. The discrete structure like the coordinates of the atoms, sources, and sinks also provides crucial information. Several works have demonstrated the rich geometric information of these data to boost the performance of other machine learning models, e.g., incorporating electron density data to better predict the physical properties of molecules [1, 22, 52].

It is common that these data themselves have inherently complicated 3D geometric structures. Work on directly predicting these structures, however, remains few. The traditional ways of obtaining such continuous data often rely on quantum chemical computation as the approximation method to solve ODEs and PDEs. For example, the ground truth electron density is often obtained with *ab initio* methods [48, 26] with accurate results but an $N^7$ computational scaling, making it prohibitive or inefficient for large molecules. Other methods like the Kohn-Sham density functional theory (KS-DFT) [24] has an $N^3$ computational scaling with a relatively large error. Therefore, building an efficient and accurate machine learning-based electron density estimator will have a positive impact on this realm.

Similar to the crucial concept of *equivariance* for discrete 3D scenarios, we can also define equivariance for a function defined on $\mathbb{R}^3$ as the property that the output transforms in accordance with the transformation on the input data. The equivariance property demonstrates the robustness of the model in the sense that it is independent of the poses of the input structure, thus also serving as an implicit way of data augmentation such that the model is trained on the whole trajectory of the input sample. Equivariance on point clouds can be obtained with vector neuron-based models [6, 16, 45, 15] and

37th Conference on Neural Information Processing Systems (NeurIPS 2023).

tensor field networks [50, 10]. We notice the close relationship between the tensor field network (TFN) and the equivariance of the continuous functions and also propose our equivariant architecture based on the tensor product.

In this way, we define our task as equivariant neural operator learning. We roughly summarize previous work on operator learning into the following four classes: 1) voxel-based regression (3D-CNN) [47, 40, 4]; 2) coefficient learning with a pre-defined set of basis functions [2, 27, 49, 14]; 3) coordinate-based interpolation neural networks [17, 18]; and 4) neural operator learning [28, 25, 29, 32]. The voxel-based 3D-CNN models are straightforward methods for discretizing the continuous input and output but are also sensitive to the specific discretization [25]. The coefficient learning models provide an alternative to discretization and are invariant to grid discretization. However, as the dimension of the Hilbert space is infinite, this method will inevitably incur errors with a finite set of basis functions. The coordinate-based networks take the raw coordinates as input and use a learnable model to "interpolate" them to obtain the coordinate-specific output. They leverage the discrete structure and provide a strong baseline, but a hard cut-off distance prevents long-distance interaction. The neural operators (NOs) are the newly-emerged architecture specifically tailored for operator learning with strong theoretical bases [28]. However, current NOs are mostly tested only on 1D or 2D data and have difficulty scaling up to large 3D voxels. They also ignore the discrete structure which provides crucial information in many scenarios.

To leverage the advantages of these methods while mitigating their drawbacks, we build our model upon the coefficient learning framework with an additional equivariant residual operator layer that finetunes the final prediction with the coordinate-specific information. A graphical overview of our model architecture is shown in Fig.1. We also provide a theoretical interpretation of the proposed neural operator learning scheme from the graph spectrum view. Similar to its discrete counterpart of graph convolutional network, our proposed model can be viewed as applying the transformation to the spectrum of the continuous feature function, thus can be interpreted as the spectral convolution on a *graphon*, a dense graph with infinitely many and continuously indexable nodes. In this way, we term our proposed model "InfGCN". Our model is able to achieve state-of-the-art performance across several large-scale electron density datasets. Ablation studies were also carried out to further demonstrate the effectiveness of the architecture.

To summarize, our contributions are, 1) we proposed a novel architecture that combines coefficient learning with the coordinate-based residual operator layer, with our model guaranteed to preserve SE(3)-equivariance by design; 2) we provided a theoretical interpretation of our model from the graph spectrum point of view as graphon convolution; and 3) we carried out extensive experiments and ablation studies on large-scale electron density datasets to demonstrate the effectiveness of our proposed model. Our code is publicly available at `https://github.com/ccr-cheng/InfGCN-pytorch`.

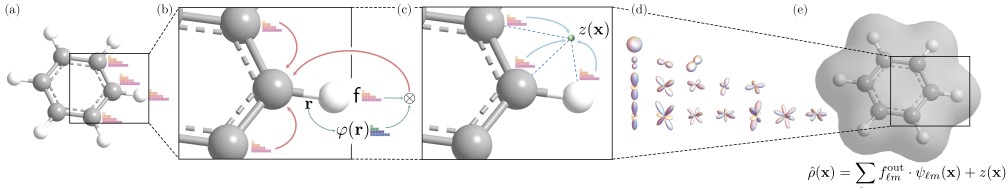

$$\hat{\rho}(\mathbf{x}) = \sum_{\ell m} f_{\ell m}^{\text{out}} \cdot \psi_{\ell m}(\mathbf{x}) + z(\mathbf{x})$$

Figure 1: Overview of the model architecture. (a) The input molecule with node-wise spherical tensor features. (b) The message passing scheme in InfGCN. $\otimes$ denotes the tensor product of two spherical tensors $\mathsf{f}, \varphi(\mathbf{r})$ (Sec.3.3). (c) Coordinate-specific residual operator layer (Sec.3.4). (d) Spherical harmonics. (e) The final prediction combining the expanded basis functions and the residue.

## 2   Preliminary

We use $\mathcal{G} = (\mathcal{V}, \mathcal{E})$ to denote the (discrete) graph with the corresponding node coordinates $\{\mathbf{x}_i\}_{i=1}^{|\mathcal{V}|}$. A continuous function over the region $\mathbb{D}$ is also provided as the target feature: $\rho : \mathcal{D} \to \mathbb{R}$. We also assume that there is an initial feature function $f_{\text{in}} : \mathcal{D} \to \mathbb{R}$ either obtained from less accurate methods, a random guess, or some learnable initialization. Formally, given $f_{\text{in}} \in L^2(\mathcal{D})$, which is a square-integrable input feature function over $\mathcal{D}$, and the target feature function $\rho \in L^2(\mathcal{D})$, we want to learn an operator in the Hilbert space $\mathcal{T} : L^2(\mathcal{D}) \to L^2(\mathcal{D})$ to approximate the target function.

Different from common regression tasks over finite-dimensional vector spaces, the Hilbert space $L^2(\mathcal{D})$ is infinite-dimensional.

## 2.1 Equivariance

Equivariance describes the behavior of the model when the input data are transformed. Formally, for group $G$ acting on $\mathcal{X}$ and group $H$ acting on $\mathcal{Y}$, for a function $f : \mathcal{X} \rightarrow \mathcal{Y}$, if there exists a homomorphism $\mathcal{S} : G \rightarrow H$ such that $f(g \cdot \mathbf{x}) = (\mathcal{S}g)f(\mathbf{x})$ holds for all $g \in G, \mathbf{x} \in X$, then $f$ is equivariant. Specifically, if $\mathcal{S} : g \mapsto e$ maps every group action to the identity action, we have the definition of *invariance*: $f(g \cdot \mathbf{x}) = f(\mathbf{x}), \forall g \in G, \mathbf{x} \in X$.

In this work, we will mainly focus on the 3D Euclidean space with the special Euclidean group SE(3), the group of all rigid transformations. As translation equivariance can be trivially achieved by using only the relative displacement vectors, we usually ignore it in our discussion and focus on rotation (i.e., SO(3)) equivariance. We first define the rotation of a continuous function $f \in L^2(\mathbb{R}^3)$ as $(\mathcal{R}f)(\mathbf{x}) := f(R^{-1}\mathbf{x})$, where $R$ is the rotation matrix associated with the rotation operator $\mathcal{R}$. Note that the inverse occurs because we are rotating the coordinate frame instead of the coordinates. In this way, the equivariance condition of an operator $\mathcal{T}$ with respect to rotation can be formulated as

$$\mathcal{T}(\mathcal{R}f) = \mathcal{R}(\mathcal{T}f), \forall \mathcal{R} \tag{1}$$

For clarity, we will distinguish equivariance and invariance, and use equivariance for functions satisfying Eq.(1).

# 3 Method

## 3.1 Intuition

Intuitively, we would like to follow the message passing paradigm [12] to aggregate information from every other point $\mathbf{x} \in \mathcal{D}$. In our scenario, however, as the nodes indexed by the coordinate $\mathbf{x} \in \mathcal{D}$ are infinite and continuous, the aggregation of the messages must be expressed as an integral:

$$\mathcal{T}_W f(\mathbf{x}) := \int_{\mathcal{D}} W(\mathbf{x}, \mathbf{y})f(\mathbf{y})d\mathbf{y} \tag{2}$$

where $W : \mathcal{D} \times \mathcal{D} \rightarrow [0, 1]$ is a square-integrable kernel function that parameterizes the source node features $f(\mathbf{y})$. There are two major problems regarding the formulation in Eq.(2): 1) unlike the discrete counterpart in which the number of nodes is finite, parameterization of the kernel $W$ in the continuous setting is hard; and 2) even $W$ is well-defined, the integral is generally intractable. Some NOs [29, 32] directly approximate the integral with Monte Carlo estimation over all grid points, which makes it harder to scale to voxels. Instead, we follow a similar idea in the coefficient learning methods [2, 27] to define a set of complete basis functions $\{\psi_k(\mathbf{x})\}_{k=1}^{\infty}$ over $L^2(\mathcal{D})$. In this way, the feature function can be *expanded* onto the basis as $f(\mathbf{x}) = \sum_{k=1}^{\infty} f_k \psi_k(\mathbf{x})$ where $f_k$ are the coefficients. We can then parameterize the message passing in Eq.(2) as the coefficient learning with truncation to the $N$-th basis. We call such an expansion method *unicentric* as there is only one basis set for expansion. In theory, as the size of the basis goes to infinite, the above expansion method can approximate any function $y \in L^2(\mathcal{D})$ with a diminishing error. In practice, however, using a very large number of bases is often impractical. The geometric information of the discrete graph is also not leveraged.

## 3.2 Multicentric Approximation

To address the limitation mentioned in the previous subsection, we leverage the discrete graph structure to build a *multicentric* expansion scheme. We use the node coordinates $\mathbf{r}_u$ in the discrete graph as the centers of basis sets: $\hat{\rho}(\mathbf{x}) = \sum_{u \in \mathcal{V}} \sum_{i=1}^{\infty} f_{i,u}\psi_i(\mathbf{x} - \mathbf{r}_u)$. We demonstrated in Appendix B that with some regularity and locality assumptions, the message passing in Eq.(2) can be parameterized as

$$f_{i,u} = \sum_{v \in \tilde{\mathcal{N}}(u)} \sum_{j=1}^{\infty} w_{ij} S_{ij}(\mathbf{r}_{uv}) f_{j,v} \tag{3}$$

where $S_{ij}(\mathbf{r}) = \int_{\mathcal{D}} \psi_i(\mathbf{x})\psi_j(\mathbf{x} - \mathbf{r})d\mathbf{x}$ models the interaction between the two displaced basis at centers $i, j$. The outer summation is over $\tilde{\mathcal{N}}(u) = \mathcal{N}(u) \cup \{u\}$, the set of neighboring centers of $u$ including $u$, and $w_{ij}$ are learnable parameters. Note that, once the basis functions are assigned, $S_{ij}$ only depends on $\mathbf{r}$, but it is generally hard to obtain the closed-form expressions. We can use neural nets to approximate it and coalesce the weight parameter into the nets.

The integral $S_{ij}(\mathbf{r})$ is often referred to as the *overlap integral* in quantum chemistry. The basis functions can be viewed as the atomic orbitals and, in this way, the integral can therefore be interpreted as the overlap between two displaced atom-centered electron clouds. The evaluation of the overlap integral is important in the self-consistent field method (Hartree–Fock method) [48].

### 3.3 Equivariant Message Passing

We will now consider the functions on the 3-dimensional Euclidean space, i.e., $\mathcal{D} = \mathbb{R}^3$, as they are widely encountered in practical applications and non-trivial to achieve equivariance. It is not easy to find a set of *equivariant basis* that satisfies Eq.(1). Inspired by the atomic orbitals used in quantum chemistry, we construct the basis function with a Gaussian-based radial function $R_n^\ell(r)$ and a spherical harmonics $Y_\ell^m(\hat{\mathbf{r}})$:

$$\psi_{n\ell m}(\mathbf{r}) = R_n^\ell(r)Y_\ell^m(\hat{\mathbf{r}}) = c_{n\ell}\exp(-a_n r^2)r^\ell Y_\ell^m(\hat{\mathbf{r}}) \tag{4}$$

where $r = |\mathbf{r}|$ is the vector length and $\hat{\mathbf{r}} = \mathbf{r}/r$ is the direction vector on the unit sphere. $c_{n\ell}$ are normalizing constants such that $\int_{\mathbb{R}^3} |\psi_{n\ell m}(\mathbf{r})|^2 dV = 1$. The degree of the spherical harmonics $\ell$ takes values of non-negative integers, and the order $m$ takes integers values between $-\ell$ and $\ell$ (inclusive). Therefore, there are $2\ell + 1$ spherical harmonics of degree $\ell$. In this way, the basis index $i, j$ are now triplets of $(n, \ell, m)$.

To further incorporate the directional information, we follow the Tensor Field Network [50] to achieve equivariance based on the tensor product. Note that for any index pair $(n_1\ell_1 m_1, n_2\ell_2 m_2)$, the overlap integral $S(\mathbf{r})$ can also be expanded onto the basis as $S(\mathbf{r}) = \sum_{n\ell m} s_{n\ell m}\psi_{n\ell m}(\mathbf{r}) =: \sum_{n\ell m} \varphi_{n\ell m}(\mathbf{r})$ [1]. For a fixed $\mathbf{r}$ and radial index $n$, the coefficient sequence $\varphi = \{\varphi_{\ell m} : \ell \geq 0, -\ell \leq m \leq \ell\}$ can be viewed as a *spherical tensor*. Notice that the node feature $\mathsf{f} = \{\mathbf{f}^\ell : \ell \geq 0\}$ can also be viewed as a spherical tensor. In the following discussion, we will omit the radial function index $n$ for clarity as it is independent of rotation. TFN leverages the fact that the spherical harmonics span the basis for the irreducible representations of SO(3) and the tensor product of them produces equivariant spherical tensors. The message passing scheme in TFN is defined as:

$$\mathbf{f}_u^\ell \leftarrow \sum_{v \in \tilde{\mathcal{N}}(u)} \sum_{k \geq 0} W^{\ell k}(\mathbf{x}_v - \mathbf{x}_u)\mathbf{f}_v^k, \qquad W^{\ell k}(\mathbf{r}) = \sum_{J=|k-\ell|}^{k+\ell} \varphi_J^{\ell k}(r) \sum_{m=-J}^{J} Y_J^m(\hat{\mathbf{r}})Q_{Jm}^{\ell k} \tag{5}$$

where $Q_{Jm}^{\ell k}$ is the Clebsch-Gordan matrix of shape $(2\ell + 2) \times (2k + 1)$ and $\varphi_J^{\ell k} : \mathbb{R}^+ \rightarrow \mathbb{R}$ are learnable radial nets that constitute part of the edge tensor features. A detailed deduction is provided in Appendix A. Intuitively, as TFN can achieve equivariance for spherical tensors, the output spherical tensor interpreted as the coefficients should also give an equivariant continuous function $\hat{\rho}(\mathbf{x})$. Indeed we have

**Theorem.** *Given an equivariant continuous input, the message passing defined Eq.(5) gives an equivariant output when interpreted as coefficients of the basis functions.*

A rigorous proof of rotation equivariance can be found in Appendix A. The translation equivariance also trivially holds as we only use the relative displacement vector $\mathbf{r}_{uv} = \mathbf{x}_v - \mathbf{x}_u$. The equivariance of this scheme relies on the equivariance of the input feature map $\mathsf{f}_{\text{in}}$. Note that the 0-degree features that correspond to pure Gaussians are isotropic, so we can use these features as the initial input. In practice, we use atom-specific embeddings to allow more flexibility in our model.

Also note that for $v = u$, the message passing can be simplified. As the spherical harmonics are orthogonal, the overlap integral is non-zero only if $m_1 = m_2$. Therefore,

$$\mathbf{f}_u^\ell = w^\ell \mathbf{f}_u^\ell + \sum_{v \in \mathcal{N}(u)} \sum_{k \geq 0} W^{\ell k}(\mathbf{x}_v - \mathbf{x}_u)\mathbf{f}_v^k \tag{6}$$

---

[1]If we use an infinite number of complete orthonormal basis, the expansion can be achieved without error. However, if we use some none-complete or finite basis, this should be viewed as an approximation in the subspace spanned by the basis.

The first term is referred to as *self-interaction* in previous papers [50, 44], but can be naturally inferred from our message passing scheme. For the nonlinearity, we follow [50] to use the vector norm of each degree of vector features:

$$f^0 = \sigma_0(f^0), \qquad \mathbf{f}^\ell = \sigma_\ell(\|\mathbf{f}^\ell\|_2)\mathbf{f}^\ell \tag{7}$$

where $\sigma_k$ are the activation functions. The equivariance holds as the vector norm is invariant to rotation. Also, to avoid over-parametrization and save computational resources, we only consider the interactions within the same radial index: $\hat{S}_{n\ell m, n'\ell'm'}(\mathbf{r}) := \delta_{nn'} S_{mm',\ell\ell'}(\mathbf{r})$. Note that this assumption generally does not hold even for orthogonal radial bases, but in practice, the model was still able to achieve comparable and even better results (Sec.5.3).

### 3.4 Residual Operator Layer

The dimension of the function space is infinite, but in practice, we can only use the finite approximation. Therefore, the expressiveness of the model will be limited by the number of basis functions used. Also, as the radial part of the basis in Eq.(4) is neither complete nor orthogonal, it can induce loss for the simple coefficient estimation approach. To mitigate this problem, we apply an additional layer to capture the residue at a given query point $p$ at coordinate $\mathbf{x}$. More specifically, the residual operator layer aggregates the neighboring node features to produce an invariant scalar[2] to finetune the final estimation:

$$z(\mathbf{x}) = \sum_{v \in \mathcal{N}(p)} \sum_{k \geq 0} W_{\text{res}}^k(\mathbf{x}_v - \mathbf{x})\mathbf{f}_v^k \tag{8}$$

This scheme resembles the coordinate-based interpolation nets and was proved effective in our ablation study (Sec.5.3). Therefore, the final output function is

$$\hat{\rho}(\mathbf{x}) = \sum_{n\ell m} f_{n\ell m}\psi_{n\ell m}(\mathbf{x}) + z(\mathbf{x}) \tag{9}$$

The equivariance of the residual operator layer as well as in the finite approximation case is also provided in Appendix A. The loss function can be naturally defined with respect to the norm in $L^2(\mathbb{R}^3)$ as $\mathcal{L} = \|\hat{\rho} - \rho\|_2^2 = \int_{\mathbb{R}^3} |\hat{\rho}(\mathbf{x}) - \rho(\mathbf{x})|^2 d\mathbf{x}$.

## 4 Graph Spectral View of InfGCN

Just as the Graph Convolutional Network (GCN) [21] can be interpreted as the spectral convolution of the discrete graph, we also provide an interpretation of InfGCN as the transformation on the *graphon* spectra, thus leading to a similar concept of *graphon convolution*. We will first introduce the (slightly generalized) concept of graphon. Defined on region $\mathcal{D}$, a *graphon*, also known as graph limit or graph function, is a symmetric square-integrable function:

$$W : \mathcal{D} \times \mathcal{D} \to [0, 1], \int_{\mathcal{D}^2} |W(\mathbf{x}, \mathbf{y})|^2 d\mathbf{x}d\mathbf{y} < \infty \tag{10}$$

Intuitively, the kernel $W(\mathbf{x}, \mathbf{y})$ can be viewed as the probability that an edge forms between the continuously indexable nodes $\mathbf{x}, \mathbf{y} \in \mathcal{D}$. Now, consider the operator $\mathcal{T}_W$ defined in Eq.(2). As the integral kernel $W$ is symmetric and square-integrable, we can apply the spectral theorem to conclude that the operator $\mathcal{T}_W$ it induces is a self-adjoint operator whose spectrum consists of a countable number of real-valued eigenvalues $\{\lambda_k\}_{k=1}^\infty$ with $\lambda_k \to 0$. Let $\{\phi_k\}_{k=1}^\infty$ be the eigenfunctions such that $\mathcal{T}_W\phi_k = \lambda_k\phi_k$. Similarly to the graph convolution for discrete graphs, any transformation on the eigenvalues $\mathcal{F} : \{\lambda_k\}_{k=1}^\infty \mapsto \{\mu_k\}_{k=1}^\infty$ can be viewed as the *graphon convolution* back in the spatial domain. We note that GCN uses the polynomial approach to filter the graph frequencies as $H\mathbf{x} = \sum_{k=0}^K w_k L^k\mathbf{x}$ where $w_k$ are the parameters. Define the power series of $\mathcal{T}_W$ as:

$$\mathcal{T}_W^n f(\mathbf{x}) = \mathcal{T}_W \mathcal{T}_W^{n-1} f(\mathbf{x}) = \int_{\mathcal{D}} W(\mathbf{x}, \mathbf{y})\mathcal{T}_W^{n-1} f(\mathbf{y})d\mathbf{y}, \mathcal{T}_W^0 = \mathcal{I} \tag{11}$$

where $\mathcal{I}$ is the identity mapping on $\mathcal{D}$. A graphon filter can be then defined as $\mathcal{H}f = \sum_{k=0}^\infty w_k \mathcal{T}_W^k f$. We can also follow GCN to use the Chebyshev polynomials to approximate the graphon filter $\mathcal{H}$:

$$\mathcal{H}f \approx \theta_1 f + \theta_2 \mathcal{T}_W f \tag{12}$$

---

[2]Note that for scalars, SO(3) equivariance is equivalent to invariance.

Just as the message passing on discrete graphs can be viewed as graph convolution, we point out here that any model that tries to approximate the continuous analog $\mathcal{T}_W f$ as defined in Eq.(2) can also be viewed as *graphon convolution*. This includes InfGCN, all NOs, and coefficient learning nets. A more formal statement using the graph Fourier transform (GFT) and the discrete graph spectral theory are provided in Appendix B for completeness.

Another related result was demonstrated by Tsubaki et al. [51] that the discrete graph convolution is equivalent to the linear transformation on a poor basis function set, with the hidden representation being the coefficient vectors and the weight matrix in GCN being the basis functions. As we have shown above, the same argument can be easily adapted for graphon convolution that the message passing in Eq.(6) can be also viewed as the linear combination of atomic orbitals (LCAO) [37] in traditional quantum chemistry.

Furthermore, based on Eq.(3), we can now give a more intuitive interpretation of the radial network in TFN: it captures the magnitude of the radial part of the overlap integral $S(\mathbf{r})$ of the basis in Eq.(4). From the point convolution aspect, the TFN structure can be also considered a special case of our proposed InfGCN model. The discrete input features can be regarded as the summation of Dirac measures over the node coordinates as $f_{\text{in}}(\mathbf{x}) = \sum_u f_u \delta(\mathbf{x} - \mathbf{x}_u)$.

# 5 Experiments

We carried out extensive experiments on large-scale electron density datasets to illustrate the state-of-the-art performance of our proposed InfGCN model over the current baselines. Multiple ablation studies were also carried out to demonstrate the effectiveness of the proposed architecture.

## 5.1 Datasets and Baselines

We evaluated our model on three electron density datasets. As computers cannot truly store continuous data, all datasets provide the electron density in a volumetric form on a pre-defined grid. Atom types and atom coordinates are also available as discrete features.

**QM9**. The QM9 dataset [41, 39] contains 133,885 species with up to nine heavy atoms (CONF). The density data as well as the data split come from [17, 18], which gives 123835 training samples, 50 validation samples, and 10000 testing samples.

**Cubic**. This large-scale dataset contains electron densities on 17,418 cubic inorganic materials [53]. In our experiment setting, we first filtered out the noble gas (He, Ne, Ar, Kr, Xe) and kept only the crystal structure whose primitive cell contains less than 64 atoms. This gave 16,421 remaining data points. A data split of 14421, 1000, and 1000 for train/validation/test was pre-assigned.

**MD**. The dataset contains 6 small molecules (ethanol, benzene, phenol, resorcinol, ethane, malonaldehyde) with different geometries sampled from molecular dynamics (MD). The former 4 molecules are from [1] with 1000 sampled geometries each. The latter two are from [2] with 2000 sampled geometries each. The models were trained separately for each molecule.

To evaluate the models, we followed [17] to define the *normalized mean absolute error* (NMAE) as our evaluation metrics:
$$\text{NMAE} = \frac{\int_{\mathbb{R}^3} |\hat{\rho}(\mathbf{x}) - \rho(\mathbf{x})| d\mathbf{x}}{\int_{\mathbb{R}^3} |\rho(\mathbf{x})| d\mathbf{x}} \tag{13}$$

To avoid randomness, different from the sampling evaluation scheme in [17], we did the evaluation on the partitioned mini-batches of the full density grid. Also, to demonstrate the equivariance of InfGCN, the density and atom coordinates were randomly rotated during inference for the QM9 dataset. The rotated density was sampled with trilinear interpolation from the original grid. Equivariance is trivial for crystal data, as there is a canonical way of assigning the primitive cell. Similarly, for the MD dataset, the authors described canonical ways to align the molecules [1, 2], so we also did not rotate them. More details regarding the datasets can be found in Appendix C.

We compared our proposed model with a wide range of different baselines including **CNN** [40, 4], interpolation networks (**DeepDFT** [17], **DeepDFT2** [18], **EGNN** [43], **DimeNet** [23], **DimeNet++** [11]), and neural operators (**GNO** [28], **FNO** [29], **LNO** [32]). For InfGCN, we set the maximal degree of spherical tensors to $L = 7$, with 16 radial basis and 3 convolution layers. For CNN and

neural operators, an atom type-specific initial density function is constructed. A sampling scheme is used for all models except for CNN. All models were trained on a single NVIDIA A100 GPU. More specifications on the model architecture and the training procedure can be found in Appendix D.

## 5.2 Main Results

The NMAE of different models on various datasets are shown in Table 1. Models with the best performance are highlighted in bold. Our model is able to achieve state-of-the-art performance for almost all datasets. For QM9 and Cubic, the performance improvement is about 1% compared to the second-best model on both rotated and unrotated data, which is significant considering the small loss. We also noticed that CNN worked well on small molecules in MD and QM9, but quickly ran out of memory for larger Cubic data samples with even a batch size of 1 (marked "OOM" in the table). This is because CNN ran on the full grid with a maximal number of $384^3$ voxels.

Table 1: NMAE (%) on QM9, Cubic, and MD datasets.

| Dataset/Model | | InfGCN | CNN | Interpolation Net | | | | | Neural Operator | | |
|---|---|---|---|---|---|---|---|---|---|---|---|
| | | | | DeepDFT | DeepDFT2 | EGNN | DimeNet | DimeNet++ | GNO | FNO | LNO |
| QM9 | rotated | **4.73** | 5.89 | 5.87 | 4.98 | 12.13 | 12.98 | 12.75 | 46.90 | 33.25 | 24.13 |
| | unrotated | **0.93** | 2.01 | 2.95 | 1.03 | 11.92 | 11.97 | 11.69 | 40.86 | 28.83 | 26.14 |
| Cubic | | **8.98** | OOM | 14.08 | 10.37 | 11.74 | 12.51 | 12.18 | 53.55 | 48.08 | 46.33 |
| MD | ethanol | 8.43 | 13.97 | **7.34** | 8.83 | 13.90 | 13.99 | 14.24 | 82.35 | 31.98 | 43.17 |
| | benzene | **5.11** | 11.98 | 6.61 | 5.49 | 13.49 | 14.48 | 14.34 | 82.46 | 20.05 | 38.82 |
| | phenol | **5.51** | 11.52 | 9.09 | 7.00 | 13.59 | 12.93 | 12.99 | 66.69 | 42.98 | 60.70 |
| | resorcinol | **5.95** | 11.07 | 8.18 | 6.95 | 12.61 | 12.04 | 12.01 | 58.75 | 26.06 | 35.07 |
| | ethane | 7.01 | 14.72 | 8.31 | **6.36** | 15.17 | 13.11 | 12.95 | 71.12 | 26.31 | 77.14 |
| | malonaldehyde | **10.34** | 18.52 | 9.31 | 10.68 | 12.37 | 18.71 | 16.79 | 84.52 | 34.58 | 47.22 |

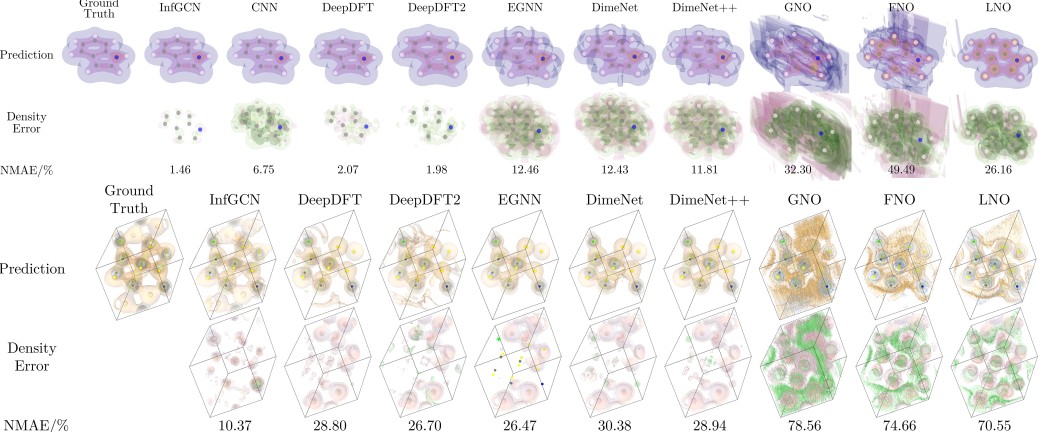

Figure 2: Visualization of the predicted density, the density error, and the NMAE. **Up**: Indole (file ID 24492 from QM9). **Down**: $Cr_4CuNiSe_8$ (mp-1226405 from Cubic). The colors of the points indicate different atom types, and the isosurfaces indicate different density values. The pink and green isosurfaces in the error plots represent the negative and positive errors, respectively.

The visualizations of the predicted densities in Fig.2 can provide more insights into the models. On QM9, the error of InfGCN had a regular spherical shape, indicating the smoothness property of the coefficient-based methods. The errors for the interpolation nets and CNN had a more complicated rugged spatial pattern. For Cubic, InfGCN was able to capture the periodicity information whereas almost all other models failed. The neural operators showed a distinct spatial pattern on the partition boundaries of the grid, as it was demonstrated to be sensitive to the partition. More visualizations are provided in Appendix E on Cubic and QM9 with a wide range of representative molecules to demonstrate the generalizability of InfGCN.

The plots of model sizes versus NMAE on the QM9 dataset are shown in Figure 3. It can be clearly seen from the figure that InfGCN achieved better performance with relatively small model size. The interpolation nets and CNN (in red) provide strong baselines. The neural operators (in orange), on the other hand, fail to scale to 3D data as a sampling scheme is required.

## 5.3 Ablation Study

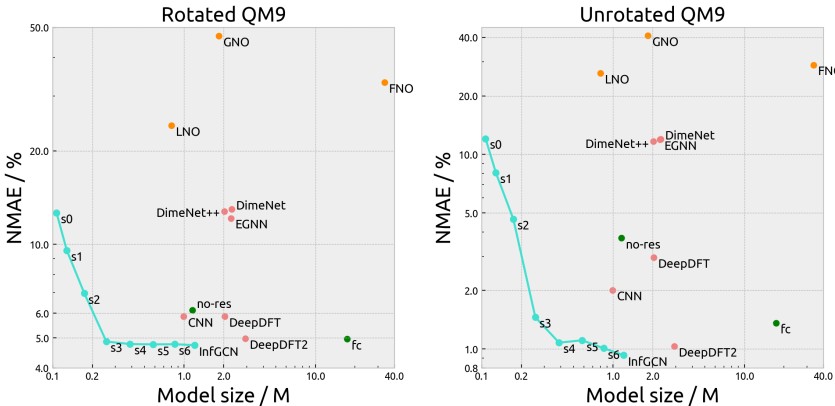

Figure 3: Plot of model sizes vs NMAE for different models and ablation studies on the QM9 dataset.

To further demonstrate the effectiveness of InfGCN, we also carried out extensive ablation studies on various aspects of the proposed architecture on the QM9 dataset. The results are summarized in Table 2 and are also demonstrated in Figure 3 in blue and green.

Table 2: NMAE (%) and the parameter count of different model settings on the QM9 dataset.

| Model | InfGCN($s_7$) | $s_6$ | $s_5$ | $s_4$ | $s_3$ | $s_2$ | $s_1$ | $s_0$ | no-res | fc |
|---|---|---|---|---|---|---|---|---|---|---|
| QM-rot (%) | **4.73** | 4.77 | 4.76 | 4.77 | 4.86 | 6.95 | 9.56 | 12.62 | 6.14 | 4.95 |
| QM-unrot (%) | **0.93** | 1.01 | 1.11 | 1.08 | 1.46 | 4.65 | 8.07 | 12.05 | 3.72 | 1.36 |
| Parameters (M) | 1.20 | 0.85 | 0.58 | 0.39 | 0.26 | 0.17 | 0.13 | 0.11 | 1.16 | 17.42 |

**Number of spherical basis.** For coefficient learning models, using more basis functions will naturally lead to a more expressive power of the model. For discrete tasks, [50, 10] used only the degree-1 spherical tensor which corresponds to vectors. We ran experiments with the maximal degree of the spherical tensor $0 \leq L \leq 7$ ($s_L$ columns). Note that $s_0$ corresponds to atom-centered Gaussian mixtures. It can be shown in Figure 3 (in blue) that the error smoothly drops as the maximal degree increases. Nonetheless, the performance gain is not significant with $L \geq 4$. This is probably because the residual operator layer can effectively finetune the finite approximation error and therefore allows for the trade-off between performance and efficiency. In this way, our proposed InfGCN can potentially scale up to larger datasets with an appropriate choice of the number of spherical basis.

**Residue prediction.** The residue prediction layer is one of the major contributions of our model that tries to mitigate the finite approximation error. It can be shown (under `no-res`) that this design significantly improves the performance by nearly 2% with negligible increases in the model size and training time. These results justify the effectiveness of the residue prediction scheme.

**Fully-connected tensor product.** As mentioned in Sec.3.3, we used a channel-wise tensor product instead a fully connected one that allows inter-channel interaction. We also tried the fully-connect tensor product under `fc`. It turns out that the fully-connected model was 15 times larger than the original model and took 2.5 times as long as the latter to train. The results, however, are even worse, probably due to overfitting on the training set.

## 6 Related Work

### 6.1 Neural Operator Learning

We use the term *neural operator* in a wider sense for any model that outputs continuous data here. For modeling 3D densities, statistical approaches are still widely used in quantum chemistry realms. For example, [2] and [49] used kernel ridge regression to determine the coefficients for atomic orbitals. [13] used a symmetry-adapted Gaussian process regression for coefficient estimation. These

traditional methods are able to produce moderate to good results but are also less flexible and difficult to scale. For machine learning-based methods, [47] utilized a voxel-based 3D convolutional net with a U-Net architecture [40] to predict density at a voxel level. Other works leveraged a similar idea of multicentric approximation. [27] and [3] all designed a tensor product-based equivariant GNN to predict the density spectra. These works are more flexible and efficient, but coefficient learning models inevitably have finite approximation errors.

Another stream of work on neural operator learning focused on directly transforming the discretized input. Tasks of these models often involve solving PDE or ODE systems in 1D or 2D scenarios. For example, [30] proposed the infinite-layer network to approximate the continuous output. Graph Neural Operator [28] approximated the operator with randomly sampled subgraphs and the message passing scheme. [29] and [9] tried to parameterize and learn the operator from the Fourier domain and spectral domain, respectively. [38] proposed an analog to the U-Net structure to achieve memory efficiency. These models are hard to scale to larger 3D data and are also sensitive to the partition of the grid if a sampling scheme is required. They also do not leverage the discrete structure.

## 6.2 Interpolation Networks

The term *interpolation network* was coined in [25] for models that take raw query coordinates as input. As graph neural networks have achieved tremendous success in discrete tasks, they are usually the base models for interpolation nets. [55] and [49] constructed the molecule graph to perform variant message passing schemes with the final query-specific prediction. [17] proposed the DeepDFT model which also considered the graph between query coordinates and [18] further extended it to use a locally equivariant GNN [45]. [20] proposed a similar model on crystalline compounds. Besides these specialized models, we also point out that current equivariant models for discrete graphs can all be adapted for continuous tasks in principle, just like DimeNet and DimeNet++ that we used as the baselines. Models that use only the invariant features including distance, angles, and dihedral angles can be trivially equivariant but lacking expressiveness [7, 44, 5, 23]. [16, 15] proposed the GVP model in which features are partitioned into scalars and vectors with carefully designed interaction to guarantee equivariance. Other works leveraged the canonical local frame [19] or tried to learn such a local frame [33]. Another line of works, the tensor field network [50, 10], utilized the group theoretical results of the irreducible representations of SO(3) and proposed a tensor product based architecture. We follow the last method as we notice the close relationship between the spherical tensor and the basis set. Though previous works with similar architecture exist [27, 3], we first give rigorous proof of the equivariance of the continuous function.

## 6.3 Downstream Applications

Several previous works tried to leverage the geometric information of the continuous function. [22] utilized the charge density and spin density as both the supervising signal and the additional input for predicting molecular energies, which achieved a significant performance improvement compared to traditional DFT-based methods. [51, 1] first projected the density onto the pre-defined basis set and then applied different neural nets on the coefficients to make predictions on downstream tasks. [46] used a 3D CNN to encode the electron density of the protein complex to predict the backbone structure. These works have demonstrated the significance of continuous data.

## 7 Limitation and Conclusion

In this paper, we introduce a novel equivariant neural operator learning architecture with the core component interpretable as the convolution on graphons. With extensive experiments, we demonstrated the effectiveness and generalizability of our model. We also discuss the limitation and potential improvement of the proposed InfGCN model in future work. As the choice of the radial basis is arbitrary, there is no theory or criterion for a better radial basis, and therefore, it leaves space for improvement. For example, we may use Slater-type orbitals (STO) instead of Gaussian-type orbitals. We may further orthogonalize the basis, which leads to the series of solutions to the Schrödinger equation for the hydrogen-like atom with more direct chemical indications. For structures with periodic boundary conditions, Fourier bases may provide a better solution. A learnable radial basis parameterized by a neural net is also a feasible option to provide more flexibility.

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

# Supplementary Material

## A    Proof of Rotation Equivariance

In this section, we will give a rigid proof of rotation equivariance of our proposed InfGCN model with finite approximation. Just as mentioned in the main text, we will ignore the radial index $n$ for clarity. Recall that we want to generate equivariant density functions as shown in Figure 4.

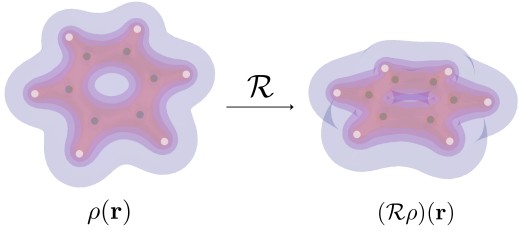

$$\rho(\mathbf{r}) \qquad\qquad (\mathcal{R}\rho)(\mathbf{r})$$

Figure 4: An illustration of rotation equivariance of the electron density function of benzene on $\mathbb{R}^3$.

**Proposition A.1.** *Rotation of a spherical harmonic of degree $\ell$ and order $m$ $(\mathcal{R}Y_\ell^m)(\mathbf{r}) := Y_\ell^m(R^{-1}\mathbf{r})$ transforms into a linear combination of spherical harmonics of the same degree:*

$$(\mathcal{R}Y_\ell^m)(\mathbf{r}) = \sum_{m'=-\ell}^{\ell} D_{mm'}^\ell(\mathcal{R})Y_\ell^{m'}(\mathbf{r}) \tag{14}$$

*where $D_{mm'}^\ell(\mathcal{R})$ is an element of the Wigner D-matrix.*

The proof of this property of spherical harmonics can be found in books on quantum mechanics, for example, Eq.4.1.4 in [8]. Therefore, for a square-integrable function defined on the unit sphere in $\mathbb{R}^3$, we can also describe the rotation of the function with Wigner D-matrics:

**Proposition A.2.** *Assume $f \in L^2(S^2)$ and the rotation of $f$ have the (infinite) expansions onto the spherical harmonic basis as:*

$$f(\mathbf{r}) = \sum_{\ell m} f_m^\ell Y_\ell^m(\mathbf{r})$$
$$\mathcal{R}f(\mathbf{r}) = \sum_{\ell m} g_m^\ell Y_\ell^m(\mathbf{r}) \tag{15}$$

*Then, we have*

$$\mathbf{g}^\ell = D_\mathcal{R}^\ell \mathbf{f}^\ell \tag{16}$$

*where $\mathbf{f}^\ell$ is the coefficient vector of degree $\ell$ with $m = 2\ell + 1$ elements, and $D_\mathcal{R}^\ell$ is the corresponding Wigner D-matrix of degree $\ell$.*

*Proof.* Notice that for each degree $\ell$, the coefficients are transformed linearly according to Eq.(14), which concludes the proof.  □

Define spherical tensor $\mathsf{f} = \{\mathbf{f}^\ell : \ell \geq 0\}$, we can further simplify the notation in Proposition A.2 as

$$\mathcal{R}f = D_\mathcal{R}(\mathsf{f}) \tag{17}$$

A pictorial illustration of the rotation of the spherical harmonics is provided in Figure 5. It can be shown that the computational diagram commutes in a sense it is equivalent to applying Wigner D-matrices on the coefficients and then projecting them back as a continuous function.

One crucial property of the spherical tensors is that the tensor product is equivariant to rotation $\mathcal{R} \in SO(3)$:

**Proposition A.3.** *The tensor product of two spherical tensors satisfies the following identity:*

$$D_\mathcal{R}(\mathsf{a} \otimes \mathsf{b}) = D_\mathcal{R}(\mathsf{a}) \otimes D_\mathcal{R}(\mathsf{b}) \tag{18}$$

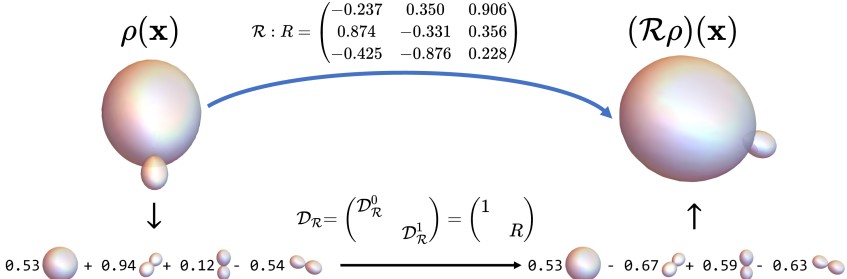

Figure 5: An illustration of rotation equivariance of linear combination of spherical harmonics as a continuous function on the unit sphere. For better visualization, the radial value used to plot the spherical harmonics is the squared density $|\rho|^2$. The calculation, however, is still done on the original (real-valued) spherical harmonics.

The proof of this property can be found in the TFN paper [50] in Appendix C. Essentially, it is a natural corollary as the property of irreducible representations. Combining Eq.(14) and (3), we can then design an equivariant message as:

$$\mathsf{f}_u = \sum_{v \in \tilde{\mathcal{N}}(u)} \sum_{\ell k} w_{\ell k} \sum_{Jm} \varphi(\mathbf{r}_{uv}) \otimes \mathsf{f}_v \tag{19}$$

where $\varphi_J^m(\mathbf{r}) = \varphi_J(r) Y_J^m(\hat{\mathbf{r}})$. Here, we use the same technique as TFN to restrict the radial functions to be independent of the order $m$ so that it is isotropic. $w_{\ell k}$ are the learnable weights. The tensor product gives a 4-dimensional tensor. The summation indices $J, m$ correspond to the two dimensions other than $\ell, k$. One way to define such a tensor product for two spherical tensors comes from the coupling of angular momentum in physics. The tensor product $\mathsf{c} = \mathsf{a} \otimes \mathsf{b}$ is defined as

$$C_{Jm} = \sum_{m_1=-\ell}^{\ell} \sum_{m_2=-k}^{k} a_{\ell m_1} b_{k m_2} \langle \ell m_1 k m_2 | Jm \rangle \tag{20}$$

where $\langle \ell m_1 k m_2 | Jm \rangle$ are the Clebsch-Gordan coefficients, and are nonzero only when $|\ell - k| \le J \le \ell + k, -J \le m \le J$. Substituting Eq.(20) into Eq.(19) and ignoring the zero terms, we will get a summation of

$$\mathbf{f}_u^\ell = \sum_{v \in \tilde{\mathcal{N}}(u)} \sum_{k \ge 0} \sum_{J=|k-\ell|}^{k+\ell} w_{\ell k} \varphi_J(r) \sum_{m=-J}^{J} Y_J^m(\hat{\mathbf{r}}) Q_{Jm}^{\ell k} \mathbf{f}_v^k \tag{21}$$

where $Q_{Jm}^{\ell k}(m_1 m_2) = \langle \ell m_1 k m_2 | Jm \rangle$. Coalescing the weight into the learnable radial function $\hat{\varphi}_J^{\ell k} = w_{\ell k} \varphi_J(r)$, we have our final message passing scheme defined in Eq.(5). With Proposition A.3, we immediately have the following corollary:

**Theorem A.4.** *The message passing defined in Eq.(19) (with infinitely many basis functions) is equivariant to rotation.*

*Proof.* According to Proposition A.2, the two spherical tensors in Eq.(19) transform as in Eq.(17). Therefore, we have

$$\mathcal{T}(\mathcal{R}\mathsf{f}_u) = \sum_{v \in \tilde{\mathcal{N}}(u)} \sum_{\ell k} w_{\ell k} \sum_{Jm} \mathcal{R}\varphi(\mathbf{r}_{uv}) \otimes \mathcal{R}\mathsf{f}_v$$

$$= \sum_{v \in \tilde{\mathcal{N}}(u)} \sum_{\ell k} w_{\ell k} \sum_{Jm} D_{\mathcal{R}}\varphi(\mathbf{r}_{uv}) \otimes D_{\mathcal{R}}\mathsf{f}_v$$

$$= \sum_{v \in \tilde{\mathcal{N}}(u)} \sum_{\ell k} w_{\ell k} \sum_{Jm} D_{\mathcal{R}}(\varphi(\mathbf{r}_{uv}) \otimes \mathsf{f}_v) \tag{22}$$

$$= D_{\mathcal{R}} \left[ \sum_{v \in \tilde{\mathcal{N}}(u)} \sum_{\ell k} w_{\ell k} \sum_{Jm} \varphi(\mathbf{r}_{uv}) \otimes \mathsf{f}_v \right]$$

$$= \mathcal{R}(\mathcal{T}\mathsf{f}_u)$$

$w_{\ell k}$ can be moved inside $D_{\mathcal{R}}$ because it is essentially a linear combination of equivariant functions, thus it is also equivariant. □

Now let's consider finite approximation where $0 \leq \ell \leq L$ for a fixed $L$. We have the following proposition:

**Proposition A.5.** *Let $\mathcal{P}_L : \mathsf{a} = \{\mathbf{a}_\ell : \ell \geq 0\} \mapsto \mathcal{P}_L \mathsf{a} = \{\mathbf{a}_\ell : 0 \leq \ell \leq L\}$ be the projection tensor operator that projects the spherical tensor onto the bases with degree less or equal to $L$, then $\mathcal{P}_L$ is rotation equivariant:*

$$\mathcal{P}_L(\mathcal{R}\mathsf{a}) = \mathcal{R}(\mathcal{P}_L\mathsf{a}) \tag{23}$$

*Proof.* It suffices to note that for each degree $\ell$, the corresponding component $\mathbf{a}^\ell$ rotates according to Eq.(17), which only depends on the components with the same degree. Therefore, $\mathcal{P}_L$ is equivariant as the components with degree $\ell \leq L$ are preserved on both sides. □

**Proposition A.6.** *The composition of two equivariant operators $\mathcal{T}_1 \circ \mathcal{T}_2$ is also equivariant.*

*Proof.*

$$(\mathcal{T}_1 \circ \mathcal{T}_2)(\mathcal{R}\mathsf{a}) = \mathcal{T}_1(\mathcal{T}_2(\mathcal{R}\mathsf{a})) = \mathcal{T}_1(\mathcal{R}(\mathcal{T}_2\mathsf{a})) = \mathcal{R}(\mathcal{T}_1\mathcal{T}_2\mathsf{a}) \tag{24}$$

□

Combining Theorem A.4 and Proposition A.5, A.6, we have the equivariant property with finite approximation:

**Corollary A.7.** *The result in Theorem A.4 also holds for a finite degree of spherical basis $0 \leq \ell \leq L$.*

Notice that for degree 0 features (scalars), equivariance is equivalent to invariance, and it can be obtained with projection operator $\mathcal{P}_0$, we have

**Corollary A.8.** *The residual operator layer defined in Eq.(8) with finite approximation is invariant with respect to the grid frame, thus equivariant to rotation under the global frame.*

Combining the results above, we immediately obtain the rotation equivariance of our proposed model.

**Theorem A.9.** *The proposed model in Eq.(9) with finite approximation and the residual operator satisfies the equivariance condition defined in Eq.(1).*

# B  Graph Spectral Theory and Graphon Convolution

In this section, we will introduce some preliminary for graph spectral theory and demonstrate the deduction for graphon convolution with more details. The basic concept of graphon can be found in various mathematics or signal processing references [36, 35, 31].

## B.1  Graph Spectral Theory

We begin with the graph spectral theory for discrete graphs. For a discrete graph $\mathcal{G} = (\mathcal{V}, \mathcal{E})$, the *graph Fourier transform* (GFT) is defined as

$$\hat{\mathbf{x}} = U^\top \mathbf{x} \tag{25}$$

where $S = U\Lambda U^\top$ is the eigenvalue decomposition of the graph shift operator $S$. A *graph shift operator* is a diagonalizable matrix $S \in \mathbb{R}^{N \times N}$ satisfying $S_{ij} = 0$ for $i \neq j, (i,j) \notin \mathcal{E}$. The graph Laplacian $L = I - A$ and the normalized version $L = I - D^{-1/2}AD^{-1/2}$ as was used in GCN [21] where $A$ is the adjacency matrix are such operators. As clear as this definition of a GFT is, it remains computationally prohibitive to implement the filtering operation on a large graph in this way. To filter the graph frequencies, a polynomial approach is thus adopted on the eigenvalues:

$$H\mathbf{x} = \sum_{k=0}^{K} w_k L^k \mathbf{x} \tag{26}$$

where $w_k$ are the learnable parameters. In the GCN formulation, the authors used the diagonalized matrix $\Lambda$ instead of $L$, which is essentially equivalent.

We now switch to the continuous *graphon* setting. As defined in Eq.(10), a graphon is a symmetric square-integrable function. The kernel $W$ induces an operator $\mathcal{T}_W$ defined in Eq.(2). As $W$ is symmetric and square-integrable, $\mathcal{T}_W$ is a self-adjoint operator that can be decomposed as

$$\mathcal{U}\mathcal{T}_W = \Lambda\mathcal{U} \tag{27}$$

where $\mathcal{U}$ is some unitary operator and $\Lambda$ is a *mulplication operator*, i.e., there exists a function $\xi(\mathbf{x})$ such that for all $f(\mathbf{x})$, $\Lambda f(\mathbf{x}) = \xi(\mathbf{x})f(\mathbf{x})$. This directly follows the result of the spectral theorem for self-adjoint operators. In this way, we may similarly define the *graphon Fourier transform* as

$$\hat{f} = \mathcal{U}f \tag{28}$$

Following the polynomial approach of approximating the graphon filter, we first define the power series of $\mathcal{T}_W$ as in Eq.(11) and use the Chebyshev polynomials of $\mathcal{T}_W$ to approximate the graphon filter $\mathcal{H}$ as $\mathcal{H}f \approx \theta_1 f + \theta_2 \mathcal{T}_W f$. Either way, parameterization and evaluation of $\mathcal{T}_W f$ is required. As mentioned in Sec.4, our model essentially operates on the eigenvalues of the operator $\mathcal{T}_W$. In the graph spectral point of view, the eigenvalues are the spectrum of the operator. Therefore, any spectral filtering can be effectively viewed as *graphon convolution*. More details regarding parameterization will be discussed below.

## B.2  Approximating Graphon Convolution

We now consider parameterization and evaluation of $\mathcal{T}_W$ to deduce Eq.3. For any complete orthonormal basis $\{\psi_k\}_{k=1}^{\infty}$ of $L^2(\mathcal{D})$, any square-integrable function $f$ can be expanded as $f = \sum_{k=1}^{\infty} f_k \psi_k$ where $f_k = \int_{\mathcal{D}} f(\mathbf{x})\psi_k(\mathbf{x})d\mathbf{x}$. We can then arrange the transform $g = \mathcal{T}_W f$ as the following matrix-vector form $\mathbf{g} = \mathbf{W}\mathbf{f}$, where

$$W_{ij} = \int_{\mathcal{D}} \psi_i(\mathbf{x}) \int_{\mathcal{D}} W(\mathbf{x}, \mathbf{y})\psi_j(\mathbf{y})d\mathbf{x}d\mathbf{y} \tag{29}$$

If $\{\psi_k\}_{k=1}^{\infty}$ coincide with the eigenfunctions $\{\phi_k\}_{k=1}^{\infty}$ which satisfy

$$\mathcal{T}_W \phi_k = \lambda_k \phi_k \tag{30}$$

We have $W_{ij} = \lambda_j \int_{\mathcal{D}} \phi_i(\mathbf{x})\phi_j(\mathbf{x})d\mathbf{x}$. For the unicentric setting, $W_{ij}$ is non-zero only when $i = j$ (the self-interaction term). For the multicentric setting, however, the computation is different. Recall that in the multicentric setting, we assume the global feature function is the summation of all atom-centered functions

$$\hat{\rho}(\mathbf{x}) = \sum_{u \in \mathcal{V}} \sum_{i=1}^{\infty} f_{i,u}\psi_i(\mathbf{x} - \mathbf{r}_u) \tag{31}$$

where $\mathbf{r}_u$ is the coordinate of center $u$. Similarly, considering one center at the origin with the other at $\mathbf{r}$, we have

$$W_{ij} = \int_{\mathcal{D}} \phi_i(\mathbf{x})\mathcal{T}_W \phi_j(\mathbf{x} - \mathbf{r})d\mathbf{x} = \lambda_j \int_{\mathcal{D}} \phi_i(\mathbf{x})\phi_j(\mathbf{x} - \mathbf{r})d\mathbf{x} \tag{32}$$

The "overlap integral" $S_{ij}(\mathbf{r})$ arises here and we further parameterize the integral with $w_{ij}$ as the basis transformation also involves index $i$. Therefore, using the above matrix-vector formulation, we have the following parameterization:

$$f_i \leftarrow \sum_{j=1}^{\infty} w_{ij}S_{ij}(\mathbf{r})f_j \tag{33}$$

If we also assume the locality of the interaction between two expansion centers, we can sum over all neighboring nodes to give the result in Eq.(3). The locality assumption often holds as the basis functions decay exponentially. Therefore, ideally, the overlap integral between two far-away centers should be negligible.

### B.3 Graphon Convolution and Continuous MPNN

Previous work regarding continuous message-passing neural networks is available. We briefly review them here and discuss their relation to our proposed method of graphon convolution. The idea of generalizing the discrete message-passing paradigm to continuous cases is essentially the same procedure as we have described in Sec.3.1 and all previous work used Eq.(2) to formulate the continuous message passing. For example, [42] proposed WNN (W refers to the graphon kernel $W$) as the limiting object GNNs with an increasing number of nodes and explored the transferability of this continuous formulation. [34] proposed cMPNN that explicitly modeled continuous functions and provided theoretical guarantees of the convergence and generalization error under discretization. [54] proposed MNN for modeling mappings between continuous manifolds and also leveraged the graph limit view of large graphs.

Though sharing a similar idea of utilizing continuous geometric structures, our method is fundamentally different from the above models. Most significantly, in the above work, the authors either explicitly constructed the graphon kernel $W$ (WNN, MNN) or directly estimated the Fredholm integral (cMPNN) in a similar fashion as various neural operators. Our approach, on the other hand, implicitly constructed the graphon and parameterized them in the spectral domain. We noted in Sec.3.1 that the kernel $W$ does not have a canonical form in our setting and Monte Carlo estimation is prohibitive for large voxels. Instead, we defined a basis set and demonstrated in previous subsections that transformation on the coefficients can be also viewed as graphon convolution in the spatial domain. In this way, we implicitly assume that there exists a different graphon for each data sample defined by their discrete structure and their categorical information. Nonetheless, the parameterization of graphon was done with the same graphon convolution for the whole dataset, as we expected this information to be generalizable across different samples. In the abovementioned work, however, a different net needs to be trained for a different graphon.

In terms of the problem formulation, we further assume there exists an underlying discrete structure that has a significant physical connotation, e.g., atoms in electron density. The datasets on electron density we experimented with are real-world data and are significantly larger than those used in previous graphon-related work. Based on the above difference in data structure and tasks, we designed a new network architecture that is different from the work before. We approximated the graphon convolution with neural nets on the coefficients instead of the feature functions themselves, and we also proposed the residual operator layer to mitigate the finite approximation error. Also, we extended the definition of rotation equivariance to continuous functions and provided rigid proof that our model achieves such a desirable property by design.

In conclusion, our work should be viewed as parallel to the existing work on graphons. We are also aware of the values of previous theoretical work. As our model still followed the framework on estimating and parameterizing the integral in Eq.2, the theoretical results on convergence and transferability could be adapted for our model to make it more concrete and solid.

## C  Datasets

In this section, we provide more details about the datasets we used in the experiments.

**QM9**. The densities are calculated with VASP using the Perdew–Burke-Ernzerhof (PBE) functional [17, 18]. The grid coordinates are guaranteed to be orthogonal but are generally different for different molecules.

**Cubic**. The densities are calculated with VASP using the projector augmented wave (PAW) method [53]. As the crystal structure satisfies the periodic boundary condition (pbc), the volumetric data are given for a primitive cell with translation periodicity. We only focus on the total charge density and ignore the spin density. Note that though all materials belong to the cubic crystal system, some of the face-center cubic (fcc) structures are given in its non-orthogonal primitive rhombohedral cell.

**MD**. Densities from [1] are calculated with the PBE XC functional; densities from [2] are calculated with Quantum ESPRESSO using the PBE functional. Both datasets are simulated in a cubic box with a length of 20 Bohr and a uniform grid size of $50^3$. The result volumetric density is represented in Fourier basis, so we first converted it into the Cartesian form for all models.

The raw data of these datasets come in different formats. We defined a unified data interface to facilitate experiments and easy extension to other datasets. A data point consists of the following data fields for training and inference:

- `atom_type`: Atom types of size $N$.
- `atom_coord`: Atom coordinates of size $(N, 3)$.
- `density`: Voxelized density value of size $N_x \times N_y \times N_z$. Note that it was flattened to have the order of X-Y-Z.
- `grid_coord`: Coordinates of the grid points where densities are sampled, with size $(N_x \times N_y \times N_z, 3)$.
- `shape`: A 3D vector representing the discretization sizes of each of the three dimensions.
- `cell`: A 3-by-3 matrix representing the cell vectors.

Other common statistics of the datasets are summarized in Table 3. The MD dataset does not have a validation split. The number of grids and grid lengths is for a single dimension so the number of voxels scales cubically with respect to it.

Table 3: Dataset details.

| Dataset | QM9 | Cubic | MD |
|---|---|---|---|
| train/val/test split | 123835/50/10000 | 14421/1000/1000 | 1000(2000)/500(400) |
| max/min/mean #grid | 160/40/87.86 | 448/32/93.97 | 20/20/20 |
| max/min/mean #node | 29/3/17.98 | 64/1/10.49 | 14/8/10.83 |
| max/min/mean length (Bohr) | 15.83/4.00/8.65 | 26.20/1.78/5.82 | 20.00/20.00/20.00 |
| #node type | 5 | 84 | 3 |

## D  Model and Training Specification

In this section, we provide our model specifications as well as the baseline model specifications. Training- and testing-related hyperparameters used in the experiments are also provided.

### D.1  Model Specification

We provide more details about the proposed InfGCN model and the baseline models. The baseline models' architectures are briefly described and major hyperparameters are provided. The model sizes provided here are for QM9.

**InfGCN**. We used spherical degree $\ell \leq 7$ and the number of radial bases $n = 16$ with the Gaussian parameters $a_k$ starting at 0.5 Bohr and ending at 5.0 Bohr. The distance was first embedded in a 64-dimensional vector and went through two fully-connected layers with a hidden size of 128. We used 3 InfGCN layers. This model has 1.20M trainable parameters and was used for all datasets.

**CNN** [40, 4]. We used a 3D-CNN with the U-Net architecture which has been successful in biomedical imaging tasks. CNN is generally not rotation equivariant. As the density grids in the datasets are not necessarily the same, we manually injected the grid information by pre-computing the initial feature map on the grid points as:

$$f_k(\mathbf{x}) = \sum_{u \in \mathcal{V}} \exp\left(-a_k \frac{|\mathbf{x} - \mathbf{x}_u|^2}{r_u}\right) \tag{34}$$

where $r_u$ is the covalent radius of atom $u$ and $\{a_k\}$ are pre-defined Gaussian parameters that contribute to the feature channel. The initial feature map was built with 16 Gaussian parameters $a_k$ starting at 0.5 Bohr and ending at 5.0 Bohr. We used a 3-level U-Net with 32, 64, and 128 feature channels, respectively. The resultant model has 990k trainable parameters.

The following 5 baselines are *interpolation nets*, as they take query coordinates and try to interpolate them from the node (atom) information.

**DeepDFT** [17] and **DeepDFT2**[18]. DeepDFT is a GNN-based network that models the interaction between the atom vertices and the query vertices for which the charge density is predicted. As DeepDFT only takes the invariant features of atom types and edge distance as input, it is also globally equivariant. DeepDFT2 uses PaiNN [45] as the GNN architecture. PaiNN designs equivariant interaction between scalar and vectorial features. Therefore, DeepDFT2 is locally equivariant. We simply followed the original model architectures which gave models of 2.04M and 2.93M trainable parameters, respectively.

**EGNN** [43]. EGNN defines an equivariant message passing based on invariant edge features like distance embedding. We used 4 layers of EGNN with an input dimension of 128 and hidden and output dimension of 256, resulting in a larger model than the original EGNN paper. We also added a query-specific residual GNN similar to InfGCN. The model has 2.27M trainable parameters.

**DimeNet** [23] and **DimeNet++** [11]. DimeNet uses spherical 2D Fourier-Bessel basis functions (2D analogs to spherical harmonics) to embed bond angles, hoping to capture geometric information about the interaction between atoms. They have achieved SOTA performance on physical property prediction. We slightly modified the original models to output a 128-dimensional feature for each atom and added a query-specific residual GNN similar to InfGCN. All the other model hyperparameters are the same as the original models. As a result, the two models have 2.31M and 2.02M parameters, respectively.

The following 3 baselines are *neural operators*. They directly try to parameterize the Fredholm operator in Eq.(2) using various approaches. Same as CNN, they cannot automatically capture the grid information, so we also use the feature in Eq.(34) as the initial feature. The initial feature map for these models is built with 32 Gaussian parameters $a_k$ starting at 0.5 Bohr and ending at 5.0 Bohr. For all NOs, a sampling training and inference scheme is utilized. We will discuss it in detail in the next subsection.

**GNO** [28]. The Graph Neural Operator (referred to as Graph Kernel Network or GKN in the original paper) tries to parameterize the Fredholm operator in Eq.(2) with the message passing on Monte Carlo sampled random subgraphs:

$$f(\mathbf{x}) \leftarrow \sigma \left( W f(\mathbf{x}) + \frac{1}{|\mathcal{N}(u)|} \sum_{\mathcal{N}(u)} \mathcal{F}(\mathbf{x}, \mathbf{y}, f(\mathbf{x}), f(\mathbf{y})) f(\mathbf{y}) \right) \tag{35}$$

where $\mathcal{F}$ is a neural net. Note that GKN is neither translation equivariant nor rotation equivariant. We used a feature size of 384 and stacked 4 convolutional layers. The cut-off distance was 5.0 Bohr for all datasets. The resultant model has 1.84M trainable parameters.

**FNO** [29]. The Fourier Neural Operator does the parameterization in the Fourier domain:

$$f(\mathbf{x}) \leftarrow \sigma \left( W f(\mathbf{x}) + \mathcal{F}^{-1} R(\mathcal{F}f)(\mathbf{x}) \right) \tag{36}$$

where $\mathcal{F}, \mathcal{F}^{-1}$ are the Fourier transform and inverse Fourier transform over the sampled data points, and $W, R$ are learnable parameter matrices. For the parameterization in the Fourier domain, only a fixed number of low-frequency Fourier modes are kept for efficiency. We used a feature size of 128 with a number of Fourier modes of 128 and stacked 4 layers. The cut-off distance was 5.0 Bohr for all datasets. The resultant model has 33.63M trainable parameters.

**LNO** [32]. The Linear Neural Operator is based on the low-rank decomposition of the kernel $W(\mathbf{x}, \mathbf{y}) := \sum_{j=1}^{r} \phi_j(\mathbf{x}) \psi_j(\mathbf{y})$, similar to the unstacked DeepONet proposed in [32]. We used a feature size of 384 with a rank of 64 and stacked 4 layers. The cut-off distance was 5.0 Bohr for all datasets. The resultant model has 803k trainable parameters.

To facilitate model training and evaluation, we also defined a unified model interface such that each model takes the atom types, atom coordinates as defined above, and the sampled densities and sampled grid coordinates which we will cover below. The model outputs predicted density values at each sampled grid point.

## D.2 Training Specification

We followed [17] to use a sampling training scheme, and we also adapted it for every model except for 3D-CNN. During training and validation, only a small portion of the grid is randomly sampled

with the corresponding density values. This scheme drastically reduces the required GPU memory, as there were cases when the whole voxel could not fit into a single GPU. During inference, however, all voxels were partitioned into mini-batches for a full evaluation to avoid randomness for a more convincing result. The 3D-CNN model required the whole voxel information, so the sampling scheme was not used.

As was demonstrated in [28], this sampling scheme can be best understood as Nyström approximation of the integral in Eq.(2). The original FNO and LNO models used the whole grid points for the estimation of the integral (Monte Carlo approximation). This is one of the major reasons that these models cannot scale to 3D voxel data. In our experiment, FNO and LNO would cause OOM even for the QM9 dataset with a batch size of 1.

The training and testing specifications are provided in Table 4. The cut-off in Bohr refers to the cut-off distance for building the atom-atom graph and the atom-query graph for InfGCN and interpolation nets. All training was done on a single NVIDIA A100 GPU. For efficiency, the testing for QM9 was done on the last 1600 samples, so larger molecules were tested. For Cubic and MD, testing was done on all the test samples.

Table 4: Training specifications.

| Model | Dataset | cutoff | n_iter | lr | patience | batch_size | lr_decay | train_sample | inf_sample |
|---|---|---|---|---|---|---|---|---|---|
| **InfGCN** | QM9 | 3.0 | 40k | 1e-3 | 10 | 64 | | | |
| | Cubic | 5.0 | 10k | 5e-3 | 5 | 32 | 0.5 | 1024 | 4096 |
| | MD | 3.0 | 2k | 5e-3 | 5 | 64 | | | |
| **CNN** | QM9 | NA | 100k | 3e-4 | 10 | 4 | 0.5 | NA | NA |
| | MD | | 4k | 1e-3 | 5 | 32 | | | |
| **DeepDFT/** | QM9 | 3.0 | 40k | 3e-4 | 10 | 64 | | | |
| **DeepDFT2** | Cubic | 5.0 | 10k | 3e-4 | 10 | 32 | 0.5 | 1024 | 4096 |
| | MD | 3.0 | 2k | 1e-3 | 5 | 64 | | | |
| **EGNN/** | QM9 | 5.0 | 40k | 3e-4 | 10 | | | | |
| **DimeNet/** | Cubic | 5.0 | 10k | 3e-4 | 10 | 64 | 0.5 | 1024 | 4096 |
| **DimeNet++** | MD | 3.0 | 2k | 1e-3 | 5 | | | | |
| **GNO/** | QM9 | | 80k | 3e-4 | 10 | | | | |
| **FNO/** | Cubic | NA | 10k | 3e-4 | 10 | 32 | 0.5 | 1024 | 4096 |
| **LNO** | MD | | 2k | 1e-3 | 5 | | | | |

## D.3 Complexity Analysis

The naïve implementation of a message-passing layer with padding and masking scales to $O(|\mathcal{E}|C(\ell+1)^6)$ where $|\mathcal{E}|$ is the number of edges and $C$ is the number of channels. This is because the message passing step involves a tensor product of two spherical tensors and the Clebsch-Gordan coefficients have their six indices all summed. However, note that there are only $(\ell+1)^2$ spherical harmonics with the degree up to $\ell$. If coefficients are compressed into one long vector, the complexity can be reduced to $O(|\mathcal{E}|C(\ell+1)^4)$. During the expansion of the basis functions on the voxels, the time complexity is $O(|\mathcal{E}|KC(\ell+1)^2)$ where $K$ is the number of voxels sampled. In practice, we noticed that a small $\ell$ suffices so that $(\ell+1)^4$ can be viewed as a constant ($\ell=7$ corresponds to 4096). Also, the Clebsch-Gordan coefficients can be pre-computed and stored, and the summation can be efficiently done by index manipulation. Our implementation was based on the e3nn[3] package which implements efficient spherical vector manipulations.

In comparison, most GNN-based layers scale as $O(|\mathcal{E}|D^2)$ where $D$ is the hidden feature size. Therefore, in our standard-setting ($C=16, \ell=7$), the time complexity is approximately of the same order (with $D=256$). For GNO, FNO, and LNO, one layer scales as $O(KD^2), O(KD^3), O(KD^2R)$, respectively. The additional coefficients are for the Fourier modes or the rank. For 3D-CNN, the time complexity scales as $O(C_{\text{in}}C_{\text{out}}N_xN_yN_zk^3)$ where $k$ is the kernel size. This is significantly larger than any of the GNN-based methods, as the whole voxel needs to be encoded and decoded. In practice, we found the interpolation nets ran slightly quicker than InfGCN and NOs, but our proposed InfGCN was able to achieve better performance.

---

[3] https://e3nn.org/

# E   Additional Results

In this section, we provide more visualization results on the QM9 and Cubic datasets to further study the generalizability of InfGCN. For the Cubic dataset, we further provide a sample on the cubic primitive cell in Figure 6.

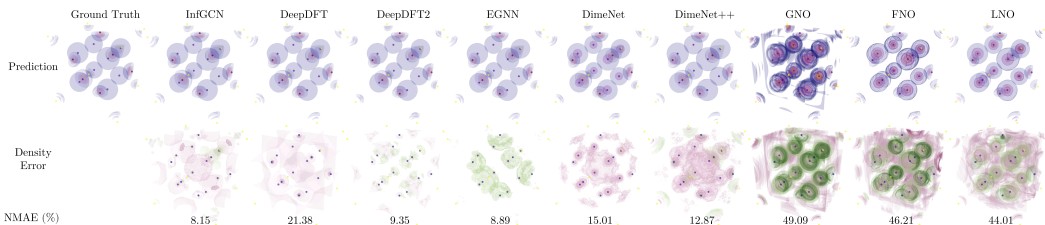

Figure 6: Visualization of the predicted density, the density error, and the NMAE for $Rb_8AsO_3$ (mp-1103058 in Cubic).

For the density visualization, we used linearly spaced values from 0.05 to 3.5 with 5 isosurfaces for the ground truth density and -0.03 (deep pink), -0.01, 0.01, and 0.03 (deep green) for the density errors for QM9 in Figure 2. We used values of 0.3, 1, 3, and 8 for the ground truth density and -0.3, -0.1, 0.1, and 0.3 for the density errors for the rhombic primitive cell in Figure 2. We used linearly spaced values from 0.5 to 8.0 with 5 isosurfaces for the ground truth density and -0.3, -0.1, 0.1, and 0.3 for the density errors for Cubic in Figure 6.

As QM9 covers a broad range of different types of molecules, we manually picked some representative molecules and evaluated all the models. The results are provided in Table 5. In order to provide a finer-grained comparison between different models, we used finer-grained isosurfaces with values of -0.03, -0.01, 0.01, and 0.03, respectively. The corresponding NMAE (%) is also provided below the plot. Molecule names, their corresponding file IDs, chemical structures, and ground truth densities are also provided in the table.

The selected molecules cover a variety of chemical types, ranging from alkane, alcohol, and ester to aromatic heterocyclic compounds. InfGCN has a significant advantage over other baselines, demonstrating its generalizability across different molecules. We also observe some patterns of the density estimation error:

- All models performed better on alkanes or the alkyl part of the molecules, e.g., the linear nonane, the branched $t$-butyl group, and the cyclic cyclohexane. One exception is cubane, which adopts an unusually sharp 90° bonding angle that would be highly strained as compared to the 109.45° angle of a tetrahedral carbon. In this example, InfGCN was still able to make a relatively accurate prediction.

- The predictions were worse for atoms with high electronegativity (F, O, N), even if the errors are normalized by the number of electrons. From a chemical aspect, a higher density will lead to a better "polarizing" ability to distort the electron cloud of the covalent atom, leading to more complicated higher-order interactions between atomic orbitals [4]. For example, the carbon atom in $CF_4$ has a significant positive partial charge, but DeepDFT overestimated its density (in pink). Noticeably, InfGCN can estimate the oxygen density with great accuracy, e.g., in glycerol, di-$t$-butyl ether, and isoamyl acetate.

- The predictions were even worse for a conjugated and aromatic system where electrons are highly delocalized[5]. Delocalization of electrons allows for long-distance interactions which are harder to capture. The presence of electron-donating groups (EDGs) and electron-withdrawing groups (EWGs) contributes greatly to the conjugated system. For example,

---

[4]The electron cloud actually won't undergo any polarizing or hybridization procedure, but it is still beneficial to think of it in this way, as it provides an intuitive interpretation of the linear combination of atomic orbitals (LCAO). In practice, such ideas are still widely used in chemistry.

[5]The term *delocalized* is also not accurate for describing electron density. However, as mentioned above, it is a convenient way of building molecular orbitals from atomic orbitals, and so are the concepts of EDG and EWG mentioned in the following text.

Figure 7: Resonance forms of aniline and nitrobenzene with formal charges.

the amino group $-NH_2$ is an EDG when bonding to a conjugated system like benzene, facilitating *ortho-* and *para-*electrophilic reactions. In contrast, the nitro group $-NO_2$ is an EWG that facilitates *ortho-* and *para-*nucleophilic reactions (See Fig.7). It can be seen from the visualization that the *ortho-* and *para-*positions of aniline are underestimated (in green) and those of nitrobenzene are overestimated (in pink) with DeepDFT and other models. For cytosine and fluorouracil, the amide (lactam) tautomeric form predominates at pH 7, further making the electron structures more complicated to achieve accurate predictions. More examples of the conjugated systems include the nitro group itself where the density of oxygen is overestimated and the amide group in asparagine where the density of the amide oxygen is underestimated and that of nitrogen is overestimated.

Table 5: More results on QM9.

| Name | File ID | Structure | Ground Truth | InfGCN | CNN | DeepDFT | DeepDFT2 | EGNN | DimeNet | DimeNet++ | GNO | FNO | LNO |
|---|---|---|---|---|---|---|---|---|---|---|---|---|---|
| ammonia | 2 | $NH_3$ | | 3.16 | 22.79 | 5.34 | 2.02 | 12.27 | 11.45 | 15.22 | 57.87 | 157.19 | 36.00 |
| urea | 20 | | | 0.99 | 3.56 | 3.33 | 1.24 | 9.63 | 9.83 | 9.94 | 44.72 | 29.73 | 22.31 |
| acetone oxime | 49 | | | 0.92 | 2.72 | 2.90 | 1.02 | 12.54 | 13.04 | 12.60 | 31.79 | 30.03 | 21.52 |
| furan | 52 | | | 1.40 | 3.81 | 2.86 | 1.68 | 12.05 | 12.51 | 12.38 | 30.68 | 39.98 | 25.38 |
| tetrafluoro-methane | 184 | $CF_4$ | | 1.76 | 4.83 | 1.95 | 1.00 | 9.79 | 6.48 | 7.75 | 34.19 | 36.31 | 27.36 |
| glycerol | 397 | | | 0.79 | 2.52 | 2.77 | 0.88 | 11.73 | 11.36 | 11.03 | 33.05 | 29.21 | 23.78 |
| cyclohexane | 658 | | | 0.54 | 1.95 | 1.75 | 0.78 | 13.21 | 12.65 | 12.60 | 30.48 | 30.80 | 19.33 |

| aniline | 940 | | | | | | | | | | | | |
|---|---|---|---|---|---|---|---|---|---|---|---|---|---|
| | | | | 1.22 | 4.67 | 2.85 | 1.30 | 12.62 | 12.80 | 12.16 | 35.29 | 31.35 | 24.60 |
| cytosine | 4318 | | | 1.25 | 5.62 | 2.96 | 1.37 | 10.35 | 10.18 | 10.26 | 40.31 | 24.58 | 20.80 |
| cubane | 19116 | | | 0.74 | 8.22 | 2.76 | 1.27 | 11.82 | 13.03 | 12.31 | 28.61 | 32.88 | 23.46 |
| purine | 24537 | | | 1.51 | 16.34 | 2.90 | 1.44 | 11.41 | 11.25 | 10.98 | 25.98 | 22.07 | 20.45 |
| di-*t*-butyl ether | 57520 | | | 0.78 | 2.28 | 2.40 | 1.04 | 12.37 | 12.79 | 11.88 | 55.77 | 27.71 | 29.73 |
| isoamyl acetate | 60424 | | | 0.67 | 2.29 | 2.67 | 0.87 | 12.41 | 12.13 | 11.65 | 82.51 | 27.16 | 30.92 |
| asparagine | 61439 | | | 1.02 | 3.71 | 3.12 | 1.11 | 10.25 | 10.05 | 10.10 | 44.62 | 25.34 | 20.53 |
| nonane | 114514 | | | 0.65 | 2.54 | 2.38 | 0.85 | 12.86 | 12.53 | 12.41 | 37.23 | 26.02 | 20.92 |
| nitrobenzene | 131915 | | | 1.44 | 22.93 | 3.41 | 1.45 | 11.00 | 10.87 | 11.16 | 47.66 | 31.38 | 22.41 |
| fluorouracil | 132701 | | | 0.98 | 4.59 | 2.61 | 0.97 | 10.06 | 9.97 | 9.90 | 51.36 | 23.77 | 16.10 |

