# OpenReview forum: "Equivariant Neural Operator Learning with Graphon Convolution"
_NeurIPS.cc/2023/Conference — NeurIPS 2023 spotlight_

### Official Review · Reviewer_SJUC · 2023-06-25

**Soundness:** 2 fair
**Presentation:** 1 poor
**Contribution:** 3 good
**Rating:** 5
**Confidence:** 3

**Summary:**

This paper proposes an architecture for learning mappings between signals/functions, which is equivariant to rigid body transformations. The architecture is based on a spectral graph convolutional network based on a novel design of a graphon/graph shift operator (or sequence of graphons/graph shift operators).

**Strengths:**

The construction of the graphon/graph shift operator underlying the spectral graph convolutional network is motivated by geometric and physical considerations. The method seems to achieve good results in experiments.

**Weaknesses:**

The construction of spectral convolution for graphons is well known, and not a contribution of this paper. See for example works by Alejandro Ribeiro, Luana Ruiz, Ron Levie, Gitta Kutyniok, and Nicolas Keriven.

The exposition and mathematical formulations are often not clear and inaccurate. It is hence hard to follow the paper. For example, the description about the construction of the graphon is not clear. Is there a single graphon, or is there a different graphon for each pair of basis elements? Moreover, the discussion around bases lacks mathematical rigor.
There seems to be a confusion between SE(3) and SO(3). Unless I am missing something, the theorems show equivariance with respect to SO(3), but the abstract and introduction talk about SE(3).
See more details in “questions.”

The exposition of the method and its motivation could be explained better. What the paper does is define a good graphon (or sequence of graphons) for problems (primarily) in quantum chemistry, and uses this graphon (or this sequence of graphons) in spectral graph convolutional networks.

It seems like this paper has potential, but it is not ready for publication at its current state.



**Questions:**

Paragraph of line 88: Note that SE(3) also includes translations, not only rotations. Perhaps you mean to develop the theory for SO(3)?

Line 99 : “the message passing must be expressed as an integral”. You define the aggregation here, not a full message passing layer. There are already papers that defined such a graphon aggregation, as I wrote above.

Line 104: “approximate the integral with Monte Carlo estimation over all grid points” - Monte Carlo methods are based on random points, not grid points, and are not affected by the curse of dimensionality directly.

Line 105: “Instead, we follow a similar idea in the coefficient learning … to define a set of complete basis functions” – basis expansion does suffer from the curse of dimensionality. So it seems like the motivating text is misleading.

Line 115: “To address the limitation mentioned in the previous subsection, we leverage the discrete graph structure to build…” Which graph? You did not discuss a graph in connection to the domain D until now.
You did not discuss the setting properly. What is the data structure/discretization that you work with?

Equation (4): In what sense is this a basis of the Hilbert space $L^2(R^3)$? It’s not an orthogonal basis or a frame of $L^2(R^3)$. It is also not a Hamel basis. It should be written more rigorously what type of expansions you consider. Perhaps instead of the term “basis” you should say a complete system (if you use a complete system), a representation system, or series expansion.
Moreover, in line 132 you write “equivariant basis”. In what sense is it equivariant and with respect to which group representation? In the text before, you discussed SE(3), but spherical harmonics have nice equivariance properties with respect to SO(3), not SE(3). You should define what you mean by equivariant basis, probably meaning that the change of basis operator is an intertwining operator of two representations of SO(3).

Line 142: What is S? Did you forget the pair of indices? In what sense can S be written in the representation system of (4)? This is not trivial as (4) is not an orthonormal basis or a frame. Perhaps you could show that (4) is complete (?), and then you can approximate S using a finite sum of elements of (4), but can you get an exact equation with a series expansion? This should be shown or explained.

Line 146: spherical harmonics are not irreducible representations. A representation is a mapping from a group to operators. The spherical harmonics provide a basis that can span the invariant subspaces of the rotation representation of SO(3).

Formula (5): you wrote an implicit equation, which is probably not what you meant. The left-hand-side should be the output feature, which is different from the input feature.

Formula (5): you are using notations that you have not defined. What is $\phi_J^{lk}$? Note that before, you missed the indices of S, so $\phi$ should have more indices.
Where is the learnable part of this message passing scheme? This seems like a definition of aggregation or shift operator.

Section 4: spectral graphon GNN is a well-known construction. Moreover, the model is not clear. Is there a single graphon underlying the method, or is there a different graphon for each pair of basis elements?


**Limitations:**

Limitations are discussed.

---

> ### Author Rebuttal · Authors · 2023-08-03
>
> # Rebuttal to Reviewer SJUC
>
> We appreciate your recognition of our work's **underlying physical connection** as well as our **comprehensive experiments**. We are willing to address your concerns and questions. Ambiguity regarding SE(3)/SO(3) is addressed in the **rebuttal to all reviewers**.
>
>
>
> ## Q1 Basis function
>
> In our setting, the basis was **pre-defined for the whole dataset** and was not a learnable part of the model. It can be shown that the basis set defined in Eq.4 indeed spans a subspace in $L^2(\mathbb{R}^3)$, as the functions are square-integrable over $\mathbb{R}^3$ and any finite set of them is linearly independent. We are aware that, however, they are neither orthogonal nor complete, as we already mentioned on line 172. We proposed multicentric approximation (Sec.3.2) and the residual operator layer to address this problem (Sec.3.4).
>
> We explained the **basis expansion** in this sense in the **rebuttal to all reviewers**, and we further explain the "**equivariant basis**" in line 132. Your understanding was correct that we want to find a basis set such that the rotation operator commutes with the neural nets on the coefficients. We will rephrase the sentence to avoid confusion.
>
>
>
> ## Q2 Construction of graphon
>
> We appreciate that you mentioned other work on spectral convolution on graphon, e.g., **WNN** (Ruiz, NeurIPS 2020), **cMPNN** (Maskey, NeurIPS 2022), and **MNN** (Wang, ACSSC 2022). These papers provided additional insights into the problem formulation. Though sharing similar ideas of utilizing continuous geometric structures, our method is fundamentally different from these methods in the following aspects:
>
> 1. **Construction of graphon**. In the above work, the authors either explicitly construct the graphon kernel $W$(WNN, MNN) or directly estimated the Fredholm integral (cMPNN) in a similar fashion as various neural operators. Our approach, on the other hand, **implicitly constructs the graphon and parameterizes them in the spectral domain**. We mentioned in line 101-105 that $W$ does not have a canonical form in our setting and Monte Carlo estimation is prohibitive for large voxels. Instead, we defined a basis setting and demonstrated that transformation on the coefficients can be also viewed as graphon convolution equivalently (Appendix B.2).
>
>    In this way, we implicitly assume that there exists **a different graphon for each data sample** defined by their discrete structure and their categorical information. In our experiments, they are the atom coordinates and atom types. Nonetheless, the parameterization of graphon was done with the same graphon convolution for the whole dataset, as we expected this information to be generalizable across different samples.
>
> 2. **Data structure & task**. In our settings, we further assume there exists an **underlying discrete structure** that has a significant physical connotation, e.g., atoms in electron density (line 19 & 74). The datasets on electron density we experimented with are **real-world** data and are **significantly larger** than those used in previous graphon-related work.
>
> 3. **Model design**. Based on the above difference in data structure and tasks, we designed a new network architecture that is different from the work before. We approximated the graphon convolution with NNs on the coefficients instead of the feature function itself, and we also proposed the **residual operator layer** to mitigate the finite approximation error. Also, we **extended the definition of rotation-equivariance** to continuous functions and provided rigid proof that our model achieves such a desirable property by design.
>
> In conclusion, we did not claim to have invented the graphon convolution, and our reference to graphon might be better viewed as an interpretation of our proposed model from the graph spectral aspect. Nonetheless, we are aware of the values of previous work on graphon in the sense that their theoretical results on convergence and transferability can be adapted for our proposed model to make it more concrete and solid. We will briefly mention these papers in the revised manuscript.
>
>
>
> ## Q3 Other questions
>
> Questions regarding **line 88 & 142 and Eq.4** are addressed in **the rebuttal to all reviewers**. We will also follow your advice on **line 99 & 146 and Eq.5** to avoid ambiguity.
>
> - **line 104 & 105**. In this claim, the rationale can be elaborated as, 1) most current Monte Carlo-based neural nets approximate the integral **using the full (2D) grid** which will be prohibitive for 3D voxels; and 2) Monte Carlo approximation requires considerably **large samples to make an accurate estimation**, as claimed in the GNO paper (one of our baselines). As for the dimensionality problem, we wanted to **expand 2D operator learning to 3D scenarios**. In our experiment settings, for example, we used 1024 basis functions, whereas the number of voxels can easily scale up to 20M, making it impractical to use the full grid. In this sense, we claimed that coefficient-based learning is more efficient and accurate than the Monte Carlo-based method.
> - **line 115**. As we mentioned in line 19-20 and 74, the discrete graph is an **additional input to our model** (and all the baselines). We mentioned in Q2 that it was one of the aspects that distinguish our work from other graphon-based work.
> - **Eq.5**. We will use $\gets$ ("gets") to avoid ambiguity. $\varphi_{J}^{\ell k}:\mathbb{R}^+\to\mathbb{R}$ are learnable neural nets that embed the radial length $r=\|\mathbf{r}\|$. Here, the radial index is also omitted for clarity. The full deduction of Eq.5, including the learnable $\varphi_{J}^{\ell k}$ can be found in Appendix A.
>
>
>
> We value your review to help make our work more comprehensive and solid, and we hope the above clarification will properly address your concerns and problems. Please comment if you have further questions. We are more than happy to address any further concerns.

---

> > ### Comment · Reviewer_SJUC · 2023-08-10
> > **Increase rating**
> >
> > Thank you for the detailed response.
> > Conditioned on all the discussed changes being implemented in the camera ready paper, I increase my score.

---

> > > ### Author Response · Authors · 2023-08-18
> > >
> > > Dear Reviewer SJUC,
> > >
> > > We sincerely appreciate your recognition of our rebuttal and your perceptive questions regarding the presentation of the original manuscript. Once again, as we have mentioned in the rebuttal, we will adopt your insightful suggestion that helps improve the conciseness and rigor of our work in our revised manuscript.
> > >
> > > Authors.

---

### Official Review · Reviewer_cyPX · 2023-07-02

**Soundness:** 3 good
**Presentation:** 2 fair
**Contribution:** 2 fair
**Rating:** 5
**Confidence:** 2

**Summary:**

This work presents a new model architecture that combines the coefficient learning scheme with coordinate-based residual operator layer for learning mapping between continuous function in 3D Euclidean space. The authors further show the proposed model can be interpreted as graph convolution on a graphon. Experiments on various datasets show the effectiveness of the proposed model.


**Strengths:**

1. Generalizing neural operator learning to 3D Euclidean space with equivariance guarantee is an interesting and underexplored research question.
2. The authors provide theoretical explanation to the model by generalizing graph convolution to graphons.
3. The proposed method performs better than other neural operators in considered datasets.

**Weaknesses:**

1. The paper is generally hard to follow as many parts are not clearly stated or explained. For example:
    - Why the target feature function $\rho$ can be expressed as the equation in line 117? What is the rationale behind the equation?
    - Why the basis function is constructed as in equation 4? How it is connected with atomic orbitals? Is such a construction necessary for achieving equivariance and good empirical performance?
    - How is the spherical tensor in section 3.3 connected with the basis function in the construction of $S$?
    - What is the motivation of using residual operator layer? Why it can mitigate the expressivity issue?
2. Following above, the presentation could be significantly improved by giving some preliminary introductions of the key technical parts used in the method, e.g. coefficient learning, concepts and background in quantum chemistry, TFN.
3. Since the method appears to be a mixture of existing approaches from different areas, it is unclear what is the novelty of this work compared with prior art on methodology aspect.
4. The authors claim the proposed neural operator to be a general framework for learning mapping between functions but only test on electron density datasets. The chosen baselines are somewhat outdated, most mainstream equivariant GNNs e.g. [1] are not considered.

[1] E (n) equivariant graph neural networks

**Questions:**

See the weakness section.

**Limitations:**

There is no negative societal impact as far as I can tell.

---

> ### Author Rebuttal · Authors · 2023-08-08
>
> # Rebuttal to Reviewer cyPX
>
> We appreciate your recognition of our work's **novel equivariant operator learning** scheme and **rigorous mathematical formulation and proof**. We also thank you for acknowledging our **extensive experiments** on large-scale datasets. We will address your concerns and questions as follows.
>
> ## Q1 Expansion of $\rho$
>
>  We apologize that the equation on line 117 should be instead a **definition**. Here, we are aware of the crucial role of the **underlying discrete structure** that has a significant physical connotation, e.g., atoms in electron density (line 19 & 74). Therefore, we took each discrete node as the center of coefficient expansion and summed over all nodes to obtain the final predicted function $\hat{\rho}$. In this way, we may mitigate the error introduced by using a finite basis set.
>
> ## Q2 Basis set
>
> We understand your concerns regarding the choice of basis set in Eq.4, and we will elaborate on the reasons that we chose basis functions in this form.
>
> - They are **widely used in quantum chemical** calculation (known as the **Gaussian-type Orbital**, GTO). The family of solutions of the Schrödinger equation of the hydrogen-like atoms shares a very similar formula as Eq.4. We also mentioned in Sec.4 (line 204) that a claim from Tsubaki et al. can be adapted for our proposed model to be interpreted as the linear combination of atomic orbitals (LCAO).
> - In our settings, the **discrete structure** also plays a vital role in determining the feature function. Therefore, it is reasonable to assume the function is more "clustered" around the node and use the node-centered Gaussians as the radial function that decays exponentially.
> - They are relatively **easy to evaluate**. It is possible to use a complete orthonormal basis, as we mentioned in Sec.7. For example, Guseinov (Int. J. Quantum Chem., 2002) proposed such a basis for $L^2(\mathbb{R}^3)$ whose time complexity of evaluating and backpropagation makes it unsuitable and unnecessary to use them in ML scenarios.
>
> We further point out that the choice of the radial part in the basis function **does not influence the equivariance** as long as the spherical harmonics part remains. Conceptually, the radial part and the angular part are independent, with the spherical harmonics completely describing the angular part. In Sec.7, we have noted this fact and proposed **potential future work** by using a different basis set that is more task-tailored.
>
>
>
> ## Q3 Construction of $S$
>
> We point out that the overlap integral $S$ is a continuous function of $\mathbf{r}$. Therefore, we can apply the same **expansion** scheme to represent $S$ as a series of coefficients (as explained in the **rebuttal to all reviewers**). Such an expansion to a spherical tensor was primarily to leverage the fact that the **tensor product of two spherical tensors is equivariant**. Therefore, representing the overlap integral as a spherical tensor, we can design an equivariant message-passing scheme for continuous feature functions.
>
> ## Q4 Motivation of the residual operator layer
>
> The proposed residual operator layer is one of the major contributions of our work, and we will elaborate on the motivation as follow:
>
> - We can only use a **finite set of basis**, which can not be complete for the infinite-dimensional space $L^2(\mathbb{R}^3)$. This problem limits the expressiveness of the overall model, which we refer to as the "finite approximation error" in the manuscript.
> - The proposed residual operator layer resembles the "interpolation nets" and takes **query-specific coordinates** as input to produce a **per-coordinate scalar** to "finetune" the density. In this sense, this layer can approximate arbitrary residue values between the coefficient-based output and the ground truth given the model is expressive enough.
> - The experiment results have demonstrated the **effectiveness** of such a residual scheme. In our ablation studies, performance improved considerably with the residual operator layer (Tab.2, "no-res").
>
>
>
> ## Q5 Preliminary & background
>
> Thanks for your valuable advice. We will add an additional introduction to related topics like quantum chemistry and TFN to make our work more self-contained.
>
>
>
> ## Q6 Comparison with prior art
>
> We already summarized our contribution to the ML realm in line 67-72, and we will also briefly discuss the comparison of our ML-based model with traditional quantum chemical calculation. As mentioned in Sec.1 the current art are various *ab initio* methods that scale as $O(N^7)$. They are often regarded as the ground truth density provided by the dataset but take considerable computation time. Other quicker quantum chemical methods like KS-DFT have an $O(N^3)$ scaling factor but less accuracy. In the DeepDFT papers (Jørgensen, 2020; *npj Comput Mater* 2022), the authors made a systematic comparison and claimed that DeepDFT outperformed KS-DFT in terms of accuracy. As our model also outperformed DeepDFT, we may obtain the qualitative conclusion that our model also makes more accurate predictions than KS-DFT.
>
> ## Q7 Datasets & baselines
>
> We further justification for using the density datasets:
>
> - The electron density provides a fundamental description of a molecule in quantum chemistry theory. The predicted density can be further utilized to **improve the performance** on other molecule-related tasks.
> - The electron density is **widely encountered** and **large-scale real-world** datasets are available. For example, the QM9 dataset contains 134k molecules, totaling up more than 1TB of data. Therefore, we believe these large-scale datasets suffice to demonstrate the effectiveness of our proposed model.
> - Other data are less explored. For example, most current work on the neural operator is tested on generated PDE as toy data or 2D data. 3D voxelized data are even scarcer, as they require significantly larger space to store.
>
> Results of **EGNN** are provided in the **rebuttal to all reviewers**.

---

> > ### Comment · Reviewer_cyPX · 2023-08-18
> >
> > Thank you for your detailed response. While many of my concerns have been appropriately addressed, I concur with reviewer SJUC's comment that the paper's presentation could benefit from a clearer explanation of formulations, motivations, and connections to prior research. Overall, I am slightly inclined towards acceptance and will maintain my score at 5.

---

> > > ### Author Response · Authors · 2023-08-18
> > >
> > > Dear Reviewer cyPX,
> > >
> > > We appreciate your valuable review and feedback. We have thoroughly considered your concerns and carefully addressed each of your questions. As we have stated in our rebuttal, we will make modifications in the revised manuscript regarding the presentation of our work that you and Reviewer SJUC mentioned. Once again, we thank you for your insightful suggestion.
> > >
> > > Authors.

---

### Official Review · Reviewer_sQju · 2023-07-06

**Soundness:** 3 good
**Presentation:** 3 good
**Contribution:** 3 good
**Rating:** 6
**Confidence:** 4

**Summary:**

This paper presents a novel framework for learning mappings between continuous functions in the 3D Euclidean space. First, the framework combines the coefficient learning scheme and the residual operator layer to maintain the sensitivity of the model to equivariance. In addition, the proposed approach utilizes both continuous and discrete graph structures of the input data to efficiently capture geometric information. Experimental results on a large-scale electron density of datasets show that the proposed model outperforms the current state-of-the-art architectures.

**Strengths:**

This paper proposes a method to ensure that the model preserves equivariance by introducing a residual operator layer in the coefficient learning framework. The paper discusses related work, outlines the approach, presents experimental results, and concludes with the potential implications of the proposed approach. The strengths of this paper are the focus on the 3D Euclidean space problem, the innovation of the equivariance techniques, and the emphasis on discrete structural information.

In terms of contributions, this paper introduces a new equivariant neural operator learning framework for the 3D Euclidean space problem. The framework effectively combines coefficient learning and residual arithmetic to leverage the strengths and mitigate the weaknesses of existing approaches. In addition, this paper provides a detailed theoretical explanation of the proposed neural operator learning scheme from a graph spectrum view, treating the proposed model as applying the transformation to a spectrum of continuous feature function. Experimental results conducted on a widely used large-scale electron density of datasets validate the superior performance of the proposed approach.


**Weaknesses:**

Some issues are as follows:

1. Although the proposed approach performs well in this paper, further baseline methods from recent years need to be added for comparison to more fully validate the effectiveness of the proposed approach.

2. It is worth noting that the proposed method might demand substantial computational resources, and its computational complexity is not explicitly discussed. As a result, its scalability to handle very large graphs might be a concern.

3. It is also necessary to pay attention to the writing standard of the paper format. For example, the title names of sections 2.1 and 3.3 are the same and need to be corrected.


**Questions:**

see weakness

**Limitations:**

While the paper does not explicitly state the limitations of the proposed method, it is important to consider certain factors that may affect its effectiveness. Specifically, the performance of the method may vary depending on the characteristics of the particular input graph. Further research is necessary to assess the generalizability and scalability of the proposed method.

---

> ### Author Rebuttal · Authors · 2023-08-08
>
> # Rebuttal to Reviewer sQju
>
> We appreciate your recognition of our **novel design of an equivariant operator learning** scheme and **detailed theoretical interpretation**. We also thank you for acknowledging our **extensive experiments** on large-scale datasets and **superior performance** over the baselines. We will address your concerns and questions as follows.
>
>
> ## Q1 More baselines
>
> We understand your concerns regarding the baseline models, and we would like to first elaborate on our experiments and baselines to provide justification for the proposed model's effectiveness.
>
> - We have experimented with a **wide variety of baseline models** ranging from GNNs, CNNs to neural operators. Many of the baselines came from recent work and often achieved SOTA performance in their tasks, as we mentioned in the **rebuttal to all reviewers**.
> - We carried out **extensive experiments and ablation studies** on large-scale real-world electron density datasets to demonstrate the effectiveness of our model. The QM9 density dataset totals up to more than 1TB of data, and the Cubic dataset contains 84 different types of atoms. In this way, the superior performance can indeed demonstrate the generalizability of InfGCN.
>
> Additional results and details of **EGNN** as another baseline are also available in the **rebuttal to all reviewers**. In conclusion, EGNN was able to achieve better performance than DimeNet/DimeNet++ on QM9 and Cubic. However, there is still a significant gap between InfGCN. We believe this result further demonstrate the effectiveness of our proposed model.
>
>
> ## Q2 Complexity analysis
>
> We made an explicit complexity analysis in **Appendix D.3** in the original manuscript, and we will briefly summarize the content here. We are aware of the high complexity of using higher-degree spherical harmonics, which has an overall complexity of $O(|\mathcal{E}|KC(\ell+1)^4)$ (with optimization), where $\mathcal{E}$ is the edge set, $K$ is the number of the sampled grid (the same for all models except for CNN), and $C$ is the number of radial channels. We noted that in our experiment setting ($\ell=7,C=16$), this factor is of approximately the same order as GNN-based models like DeepDFT with a hidden size of 256. In practice, though, we noticed that interpolation nets are slightly quicker than InfGCN and neural operators.
>
> As InfGCN essentially constructs the graph based on the coordinates of the discrete structure, it inherits the **localization assumption** of normal GNNs. In this way, scalability to larger graphs can be naturally achieved with a reasonable cut-off distance. Furthermore, in Fig.3 and Tab.2, we noted that the performance improvement from using $L=4$ to $L=7$ is not significant as a result of using the residual operator layer. Therefore, we may use a smaller $L$ for larger graphs to accelerate the calculation without significantly damaging the overall performance.
>
>
>
> ## Q3 Formatting
>
> Thanks for pointing out this problem. We will modify the title of Sec.3.3 to "Equivariant Message Passing" to avoid confusion and ambiguity.
>
>
>
> We value your review to help make our work more comprehensive and solid, and we hope the above clarification will properly address your concerns and problems. Please comment if you have further questions. We are more than happy to address any further concerns.

---

### Official Review · Reviewer_Wmsq · 2023-07-06

**Soundness:** 3 good
**Presentation:** 4 excellent
**Contribution:** 3 good
**Rating:** 7
**Confidence:** 4

**Summary:**

This paper introduces a method for learning SE(3) equivariant continuous functions in R^3 given an approximation of a continuous input and structural information in the form of a graph. The authors represent continuous inputs using ideas from coefficient learning (i.e., learning functions as the sum of learned coefficients multiplied by elements of a predefined basis), and incorporate structural information from discrete graphs by centering the basis functions at the node coordinates. To improve the quality of the learned function a residual term inspired by coordinate-based interpolation methods is used. The authors also provide a graph spectral interpretation of their method showing that the approach resembles graphon convolution. The proposed method outperforms baselines.

**Strengths:**

* Originality: The work appears to be original. Several components of the proposed method come from existing efforts, however, the method to integrate structural information and the graph spectral interpretation appear to be new.
* Quality: The work appears to be of very good quality.
* Clarity: The paper is well written and the ideas are communicated in an accessible way.
* Significance: This paper proposes a learning based approach for electron density estimation. Traditional approaches are computationally expensive, and therefore do not scale well. The proposed approach addresses this challenge by allowing for estimation in a feedforward deep learning framework.

**Weaknesses:**

* Quality: Although the model is constructed to be SE(3) equivariant, this doesn’t seem to be reflected very well in the analysis. Can the authors explain why the error for rotated QM9 is so high relative to the unrotated version in table 1 and 2?

**Questions:**

* Questions
  * As I understand S_{ij} is given by a neural network; is there a mechanism to make sure it is close to the integral form?
* Possible Typos
  * Equation 3, change f_{i,u} to f_{u,i}
  * Equation 5, and 7, consider adding \qquad after the comma for readability
  * Fig 3. Consider using the same axis upper bound for both plots to improve readability


**Limitations:**

The authors have communicated limitations of their work

---

> ### Author Rebuttal · Authors · 2023-08-08
>
> # Rebuttal to Reviewer Wmsq
>
> We really appreciate your recognition of our **originality** in designing the novel equivariant operator learning scheme and the good **quality** of the overall work. We also thank you for acknowledging the **significance** of our work in scientific domains. We will address your concerns and questions as follows.
>
>
>
> ## Q1 SE(3) equivariance
>
> We also noticed the performance gap between the rotated and unrotated QM9 dataset, and we believe it may be attributed to the following two factors:
>
> - We used **trilinear interpolation** on grid values to obtain the rotated target function instead of recalculating them from quantum chemical methods. Therefore, the interpolation may introduce errors to the rotated ground truth target function.
> - The grids rotated outside the original cell were **omitted** and the new grids that did not have near interpolation anchors were **masked out** (instead of zero-padded). In other words, only the intersected region between the original and the rotated grids was kept. Normally, the masked grids are often close to the cell boundary with near-zero target values that are easy to predict for all models. So, after masking out, the normalized MAE (MAE divided by the summation of all grid target values) will naturally increase.
>
> We would also like to point out that the equivariance property of some baselines is more obvious to demonstrate than InfGCN. For example, DeepDFT used only the scalar features like the distances between atoms and grids and was trivially equivariant. DeepDFT2 was built upon PaiNN, an equivariant GNN that designs tensorial operations between scalars and vectors, so it was also equivariant. We analyzed the equivariance of all models in Appendix D.1. Nonetheless, there was still a **performance gap in these equivariant baselines**. Therefore, we think this gap should be attributed to the bias when we constructed the rotated dataset instead of the expressiveness of the models.
>
>
>
> ## Q2 Estimation of overlap integral $S$
>
> The learnable radial net $\varphi$ in Eq.5 is closely related to the overlap integral $S$ but is not identical to the latter (as we mentioned in Sec.4 line 211). As we need to **parameterize** each InfGCN layer with different parameters, $\varphi$ is actually the combination of $S$ and the learnable weight (see Eq.3 or Eq.21 in Appendix A). Therefore, it is unnecessary and difficult in practice to enforce a strict equation between $S$ and $\varphi$. On the other hand, our proposed model is quite stable during training. Therefore, we may assume that $\varphi$ at least captures the magnitude or other features of the overlap integral.
>
>
>
> ## Q3 Typos & formatting
>
> We appreciate you pointing out these typos and formatting problems in our paper, and we will fix them in the revised manuscript.
>
> - **Eq.3**. We will modify the notation to be consistent with line 117.
> - **Eq.5 & 7**. We will add space for readability.
> - **Fig.3**. We originally used a different scale for better visualization, as there existed a performance gap between the rotated and unrotated QM9. We will revise the figure.
>
>
>
> We value your review to help make our work more comprehensive and solid, and we hope the above clarification will properly address your concerns and problems. Please comment if you have further questions. We are more than happy to address any further concerns.

---

### Official Review · Reviewer_cHMR · 2023-07-13

**Soundness:** 3 good
**Presentation:** 3 good
**Contribution:** 3 good
**Rating:** 8
**Confidence:** 3

**Summary:**

The paper introduces an rchitecture combining a coefficient learning scheme with a residual operator layer for learning mappings between continuous functions in 3D Euclidean space designed to achieve SE(3)-equivariance. The proposed method can be seen as convolution on graphons, which are dense graphs with infinitely many nodes. They claim that their approach, called InfGCN, captures geometric information while preserving equivariance by leveraging both the continuous graphon structure and the discrete graph structure of the input data.
The authors perform experiments on large-scale electron density datasets, and they claim that the proposed model demonstrates significant improvement over current state-of-the-art architectures.

**Strengths:**

The paper is well written and the topic is very relevant. The notation is clear despite the technical nature of the work.

**Weaknesses:**

A bit more clarity in the definition of equivariant is desired. Even though SO(3) irrep representation are mentioned, it should be clearer which group is referred to, e.g., SO(3), O(3), E3, etc?

**Questions:**

- Are there any advantages for explainability from the framework?

**Limitations:**

The authors discuss the limitations of their work.

---

> ### Author Rebuttal · Authors · 2023-08-08
>
> # Rebuttal to Reviewer cHMR
>
> We really appreciate your high recognition of the high quality of our work. We will address your concerns and questions as follows.
>
>
> ## Q1 SE(3)/SO(3) equivariance
>
> Clarification regarding SE(3)/SO(3) is made in the **rebuttal to all reviewers**. We thank you for pointing this out and we will make changes to the revised manuscript to avoid potential ambiguity.
>
>
>
> ## Q2 Explainability
>
> The explainability of our proposed model may be elaborated on in the following aspects:
>
> - The designed message passing can be interpreted as the linear combination of atomic orbitals (**LCAO**), which has a close connection with quantum chemistry (Sec.4).
> - The overall architecture can be viewed as **graphon convolution**. In this sense, every grid point was able to aggregate information from all other grid points in $\mathbb{R}^3$, making the model more expressive.
> - As a model for the operator learning task, **intermediate results** from different graphon convolutional layers can also be exacted and visualized to provide more insights.
>
> Still, we note that, as a machine learning model, the detailed information flow of our model also relies on the development of the theoretical parts in ML and GNN to provide more insights into the whole model.
>
>
>
> We appreciate your review to help make our work more comprehensive and solid, and we hope the above clarification will properly address your concerns and problems. Please comment if you have further questions. We are more than happy to address any further concerns.

---

### Author Rebuttal · Authors · 2023-08-08

# Rebuttal to All Reviewers

Dear Reviewers, we would like to first express our sincere gratitude for your valuable reviews to help make our work more comprehensive and solid. We thank your recognition of the innovation of our work, and we are willing to address your concerns and questions. Here we address some **common questions** and other review-specific questions and concerns are addressed separately.



## Q1 More baselines

We notice the concern of Review sQju & cyPX regarding using more recent baselines. We would like to point out that many of our baselines are newly emerged models, often with SOTA performance in their tasks. Most noticeably, **DeepDFT2** was built upon **PaiNN** (Schütt, ICML 2021), an equivariant GNN with carefully designed tensorial operation. As demonstrated in their paper, PaiNN outperformed EGNN on QM9 and DeepDFT2 indeed provided a very strong baseline. Other models like **FNO** (Li, ICLR 2021), **GNO** (Li, 2020) are also relatively new with the claimed SOTA performance.

Nonetheless, we also ran the **EGNN** model as another baseline on all datasets, as per request by cyPX. The EGNN code was adapted from the official repo by Satorras et al. In our experiment setting, we used a larger model than the original setting for a fair comparison. We used 4 layers of EGNN layers with an input dimension of 128, hidden and output dimension of 256. This resulted in 2.27M trainable parameters. Other training hyperparameters are identical to DimeNet. The results are shown below:

| Dataset          | NMAE (%) |
| ---------------- | -------- |
| QM9 rotated      | 12.13    |
| QM9 unrotated    | 11.92    |
| Cubic            | 11.74    |
| MD-ethanol       | 13.90    |
| MD-benzene       | 13.49    |
| MD-phenol        | 13.59    |
| MD-resorcinol    | 12.61    |
| MD-ethane        | 15.17    |
| MD-malonaldehyde | 12.37    |

EGNN was able to achieve better performance than DimeNet/DimeNet++ on QM9 and Cubic. However, there is still a significant gap between InfGCN.



## Q2 Equivariance wrt SE(3)/SO(3)

We are sorry for the confusion between SE(3) and SO(3) in the manuscript, and we will clarify it here. We first point out our proposed model is **translation-invariant**, as we only use the relative displacement vector (line 155). Therefore, together with SO(3) equivariance (Appendix A), our model is SE(3) equivariant. We acknowledge that for most of the part in Sec.2-4, we referred to SO(3) (rotation) when we mentioned "equivariance" in our formulation, primarily to be on par with other classical equivariance papers including SE(3)-Transformer (Fuchs, NeurIPS 2020) and EGNN (Satorras, 2021), where they also used SE(3)/E(3) formulation but mainly dealt with SO(3)/O(3) in the context. We will explicitly mention the difference between SE(3) and SO(3) in the introduction section and use SO(3) in the method section to avoid potential ambiguity.



## Q3 Overlap integral $S$ & Expansion

We also notice some ambiguity regarding $S$ in line 142. $S$ should have 6 indices $n_1\ell_1m_1n_2\ell_2m_2$ where $n_1\ell_1m_1$ are basis indices of the first expansion center and $n_2\ell_2m_2$ of the second. We omitted them for clarity as the arguments suit for all overlap integral between any of the two basis functions. As for the **expansion** of a feature function $f$, we slightly abused the notation to indicate a set of coefficients that minimize the norm:
$$
\arg\min_{c_{n\ell m}}\left\\|f(\mathbf{x})-\sum_{n\ell m}c_{n\ell m}\psi_{n\ell m}(\mathbf{x})\right\\|_2
$$
Therefore, line 142 should be viewed as a **finite basis approximation**. We believe such an approximation was reasonable because the overlap integral also decays exponentially with respect to the distance between centers. We often wrote the expansion as $f=\mathsf{f}$, where $\mathsf{f}$ is the coefficient spherical tensor indexed by $n\ell m$. In Sec 3.3 and Appendix A, we also omitted the radius $n$ as it is independent of rotation equivariance, and further used the notation of $f=\mathsf{f}=\\{\mathbf{f}^\ell,\ell\ge 0\\}$. We will use $\approx$ and "approximate" for a more rigorous description.



We value your reviews and comments and hope the above clarification will properly address your concerns and problems. Please comment if you have further questions. We are more than happy to address any further concerns.

---

### Author Response · Authors · 2023-08-15

Dear Esteemed Reviewers,

We would like to extend our heartfelt appreciation for your time and effort spent reviewing our paper. Your perceptive comments and recommendations have significantly contributed to improving the quality and coherence of our work. We have thoroughly considered your concerns and carefully addressed each of your questions. As the rebuttal phase proceeds, we eagerly anticipate your post-rebuttal feedback. We would be grateful for your feedback on whether our responses have satisfactorily resolved your concerns.

Warm regards,
Authors.

---

### Decision · Program_Chairs · 2023-09-21

**Decision:**

Accept (spotlight)

**Comment:**

The reviewers strongly recommend acceptance. The paper provides a method to develop SE(3) neural operators to output functions in 3D. The authors show significant improvement over prior art that do not preserve SE(3).